# ON THE ORIGIN OF IMPLICIT REGULARIZATION IN STOCHASTIC GRADIENT DESCENT

**Samuel L. Smith**[1]**, Benoit Dherin**[2]**, David G. T. Barrett**[1] **and Soham De**[1]
[1]DeepMind, [2]Google
`{slsmith, dherin, barrettdavid,sohamde}@google.com`

## ABSTRACT

For infinitesimal learning rates, stochastic gradient descent (SGD) follows the path of gradient flow on the full batch loss function. However moderately large learning rates can achieve higher test accuracies, and this generalization benefit is not explained by convergence bounds, since the learning rate which maximizes test accuracy is often larger than the learning rate which minimizes training loss. To interpret this phenomenon we prove that for SGD with random shuffling, the mean SGD iterate also stays close to the path of gradient flow if the learning rate is small and finite, *but on a modified loss*. This modified loss is composed of the original loss function and an implicit regularizer, which penalizes the norms of the mini-batch gradients. Under mild assumptions, when the batch size is small the scale of the implicit regularization term is proportional to the ratio of the learning rate to the batch size. We verify empirically that explicitly including the implicit regularizer in the loss can enhance the test accuracy when the learning rate is small.

## 1 INTRODUCTION

In the limit of vanishing learning rates, stochastic gradient descent with minibatch gradients (SGD) follows the path of gradient flow on the full batch loss function (Yaida, 2019). However in deep networks, SGD often achieves higher test accuracies when the learning rate is moderately large (LeCun et al., 2012; Keskar et al., 2017). This generalization benefit is not explained by convergence rate bounds (Ma et al., 2018; Zhang et al., 2019), because it arises even for large compute budgets for which smaller learning rates often achieve lower training losses (Smith et al., 2020). Although many authors have studied this phenomenon (Jastrzębski et al., 2018; Smith & Le, 2018; Chaudhari & Soatto, 2018; Shallue et al., 2018; Park et al., 2019; Li et al., 2019; Lewkowycz et al., 2020), it remains poorly understood, and is an important open question in the theory of deep learning.

In a recent work, Barrett & Dherin (2021) analyzed the influence of finite learning rates on the iterates of gradient descent (GD). Their approach is inspired by *backward error analysis*, a method for the numerical analysis of ordinary differential equation (ODE) solvers (Hairer et al., 2006). The key insight of backward error analysis is that we can describe the bias introduced when integrating an ODE with finite step sizes by introducing an ancillary *modified flow*. This modified flow is derived to ensure that discrete iterates of the original ODE lie on the path of the continuous solution to the modified flow. Using this technique, the authors show that if the learning rate $\epsilon$ is not too large, the discrete iterates of GD lie close to the path of gradient flow on a modified loss $\widetilde{C}_{GD}(\omega) = C(\omega) + (\epsilon/4)||\nabla C(\omega)||^2$. This modified loss is composed of the original loss $C(\omega)$ and an implicit regularizer proportional to the learning rate $\epsilon$ which penalizes the euclidean norm of the gradient.

However these results only hold for full batch GD, while in practice SGD with small or moderately large batch sizes usually achieves higher test accuracies (Keskar et al., 2017; Smith et al., 2020). In this work, we devise an alternative approach to backward error analysis, which accounts for the correlations between minibatches during one epoch of training. Using this novel approach, we prove that for small finite learning rates, the mean SGD iterate after one epoch, averaged over all possible sequences of minibatches, lies close to the path of gradient flow on a second modified loss $\widetilde{C}_{SGD}(\omega)$, which we define in equation 1. This new modified loss is also composed of the full batch loss function and an implicit regularizer, however the structure of the implicit regularizers for GD and SGD differ, and their modified losses can have different local and global minima. Our analysis

therefore helps explain both why finite learning rates can aid generalization, and why SGD can achieve higher test accuracies than GD. We assume that each training example is sampled once per epoch, in line with best practice (Bottou, 2012), and we confirm empirically that explicitly including the implicit regularization term of SGD in the training loss can enhance the test accuracy when the learning rate is small. Furthermore, we prove that if the batch size is small and the gradients are sufficiently diverse, then the expected magnitude of the implicit regularization term of SGD is proportional to the ratio of the learning rate to the batch size (Goyal et al., 2017; Smith et al., 2018).

We note that many previous authors have sought to explain the generalization benefit of SGD using an analogy between SGD and stochastic differential equations (SDEs) (Mandt et al., 2017; Smith & Le, 2018; Jastrzębski et al., 2018; Chaudhari & Soatto, 2018). However this SDE analogy assumes that each minibatch is randomly sampled from the full dataset, which implies that some examples will be sampled multiple times in one epoch. Furthermore, the most common SDE analogy holds only for vanishing learning rates (Yaida, 2019) and therefore misses the generalization benefits of finite learning rates which we identify in this work. An important exception is Li et al. (2017), who applied backward error analysis to identify a modified SDE which holds when the learning rate is finite. However this work still relies on the assumption that minibatches are sampled randomly. It also focused on the convergence rate, and did not discuss the performance of SGD on the test set.

**Main Result.** We now introduce our main result. We define the cost function over parameters $\omega$ as $C(\omega) = (1/N) \sum_{j=1}^{N} C_j(\omega)$, which is the mean of the per-example costs $C_j(\omega)$, where $N$ denotes the training set size. Gradient flow follows the ODE $\dot{\omega} = -\nabla C(\omega)$, while gradient descent computes discrete updates $\omega_{i+1} = \omega_i - \epsilon \nabla C(\omega_i)$, where $\epsilon$ is the learning rate. For simplicity, we assume that the batch size $B$ perfectly splits the training set such that $N\%B = 0$, where $\%$ denotes the modulo operation, and for convenience we define the number of batches per epoch $m = N/B$. We can therefore re-write the cost function as a sum over minibatches $C(\omega) = (1/m) \sum_{k=0}^{m-1} \hat{C}_k(\omega)$, where the minibatch cost $\hat{C}_k(\omega) = (1/B) \sum_{j=kB+1}^{kB+B} C_j(\omega)$. In order to guarantee that we sample each example precisely once per epoch, we define SGD by the discrete update $\omega_{i+1} = \omega_i - \epsilon \nabla \hat{C}_{i\%m}(\omega_i)$.

Informally, our main result is as follows. After one epoch, the mean iterate of SGD with a small but finite learning rate $\epsilon$, averaged over all possible shuffles of the batch indices, stays close to the path of gradient flow on a modified loss $\dot{\omega} = -\nabla \widetilde{C}_{SGD}(\omega)$, where the modified loss $\widetilde{C}_{SGD}$ is given by:

$$\widetilde{C}_{SGD}(\omega) = C(\omega) + \frac{\epsilon}{4m} \sum_{k=0}^{m-1} ||\nabla \hat{C}_k(\omega)||^2. \tag{1}$$

We emphasize that our analysis studies the mean evolution of SGD, not the path of individual trajectories. The modified loss $\widetilde{C}_{SGD}(\omega)$ is composed of the original loss $C(\omega)$ and an implicit regularizer $C_{reg}(\omega) = (1/4m) \sum_{k=0}^{m-1} ||\nabla \hat{C}_k(\omega)||^2$. The scale of this implicit regularization term is proportional to the learning rate $\epsilon$, and it penalizes the mean squared norm of the gradient evaluated on a batch of $B$ examples. To help us compare the modified losses of GD and SGD, we can expand,

$$\widetilde{C}_{SGD}(\omega) = C(\omega) + \frac{\epsilon}{4} ||\nabla C(\omega)||^2 + \frac{\epsilon}{4m} \sum_{i=0}^{m-1} ||\nabla \hat{C}_i(\omega) - \nabla C(\omega)||^2. \tag{2}$$

We arrive at Equation 2 from Equation 1 by noting that $\sum_{i=0}^{m-1} (\nabla \hat{C}_i(\omega) - \nabla C(\omega)) = 0$. In the limit $B \to N$, we identify the modified loss of gradient descent, $\widetilde{C}_{GD} = C(\omega) + (\epsilon/4)||\nabla C(\omega)||^2$, which penalizes "sharp" regions where the norm of the full-batch gradient ($||\nabla C(\omega)||^2$) is large. However, as shown by Equation 2, the modified loss of SGD penalizes both sharp regions where the full-batch gradient is large, and also "non-uniform" regions where the norms of the errors in the minibatch gradients ($||\nabla \hat{C}(\omega) - \nabla C(\omega)||^2$) are large (Wu et al., 2018). Although global minima of $C(\omega)$ are global minima of $\widetilde{C}_{GD}(\omega)$, global minima of $C(\omega)$ may not be global (or even local) minima of $\widetilde{C}_{SGD}(\omega)$. Note however that $C(\omega)$ and $\widetilde{C}_{SGD}(\omega)$ do share the same global minima on over-parameterized models which can interpolate the training set (Ma et al., 2018). We verify in our experiments that the implicit regularizer can enhance the test accuracy of models trained with SGD.

**Paper structure.** In Section 2, we derive our main result (Equation 1), and we confirm empirically that we can close the generalization gap between small and large learning rates by including the implicit regularizer explicitly in the loss function. In Section 3, we confirm Equation 1 satisfies the linear scaling rule between learning rate and batch size (Goyal et al., 2017). In Section 4, we provide additional experiments which challenge the prevailing view that the generalization benefit of small batch SGD arises from the temperature of an associated SDE (Mandt et al., 2017; Park et al., 2019).

## 2 A BACKWARD ERROR ANALYSIS OF STOCHASTIC GRADIENT DESCENT

Backward error analysis has great potential to clarify the role of finite learning rates, and to help identify the implicit biases of different optimizers. We therefore give a detailed introduction to the core methodology in Section 2.1, before deriving our main result in Section 2.2. In Section 2.3, we confirm empirically that the implicit regularizer can enhance the test accuracy of deep networks.

### 2.1 AN INTRODUCTION TO BACKWARD ERROR ANALYSIS

In numerical analysis, we often wish to integrate ODEs of the form $\dot{\omega} = f(\omega)$. This system usually cannot be solved analytically, forcing us to simulate the continuous flow with discrete updates, like the Euler step $\omega(t + \epsilon) \approx \omega(t) + \epsilon f(\omega(t))$. However discrete updates will introduce approximation error when the step size $\epsilon$ is finite. In order to study the bias introduced by this approximation error, we assume the learning rate $\epsilon$ is relatively small, and introduce a modified flow $\dot{\omega} = \widetilde{f}(\omega)$, where,

$$\widetilde{f}(\omega) = f(\omega) + \epsilon f_1(\omega) + \epsilon^2 f_2(\omega) + \dots. \tag{3}$$

The modified flow of $\widetilde{f}(\omega)$ is equal to the original flow of $f(\omega)$ when $\epsilon \to 0$, but it differs from the original flow if $\epsilon$ is finite. The goal of backward error analysis is to choose the correction terms $f_i(\omega)$ such that the iterates obtained from discrete updates of the original flow with small finite step sizes lie on the path taken by the continuous solution to the modified flow with vanishing step sizes.

The standard derivation of backward error analysis begins by taking a Taylor expansion in $\epsilon$ of the solution to the modified flow $\omega(t + \epsilon)$. We obtain the derivatives of $\omega(t + \epsilon)$ recursively using the modified flow equation $\dot{\omega} = \widetilde{f}(\omega)$ (see Hairer et al. (2006)), and we identify the correction terms $f_i(\omega)$ by ensuring this Taylor expansion matches the discrete update (e.g., $\omega_{t+1} = \omega_t + \epsilon f(\omega_t)$) for all powers of $\epsilon$. However, this approach does not clarify *why* these correction terms arise. To build our intuition for the origin of the corrections terms, and to clarify how we might apply this analysis to SGD, we take a different approach. First, we will identify the path taken by the continuous modified flow by considering the combined influence of an infinite number of discrete steps in the limit of vanishing learning rates, and then we will compare this continuous path to either a single step of GD or a single epoch of SGD. Imagine taking $n$ Euler steps on the modified flow $\widetilde{f}(\omega)$ with step size $\alpha$,

$$
\begin{align}
\omega_{t+n} &= \omega_t + \alpha\widetilde{f}(\omega_t) + \alpha\widetilde{f}(\omega_{t+1}) + \alpha\widetilde{f}(\omega_{t+2}) + \dots \tag{4} \\
&= \omega_t + \alpha\widetilde{f}(\omega_t) + \alpha\widetilde{f}(\omega_t + \alpha\widetilde{f}(\omega_t)) + \alpha\widetilde{f}(\omega_t + \alpha\widetilde{f}(\omega_t) + \alpha\widetilde{f}(\omega_t + \alpha\widetilde{f}(\omega_t))) + \dots \tag{5} \\
&= \omega_t + n\alpha\widetilde{f}(\omega_t) + (n/2)(n-1)\alpha^2\nabla\widetilde{f}(\omega_t)\widetilde{f}(\omega_t) + O(n^3\alpha^3). \tag{6}
\end{align}
$$

We arrived at Equation 6 by taking the Taylor expansion of $\widetilde{f}$ and then counting the number of terms of type $\nabla\widetilde{f}(\omega_t)\widetilde{f}(\omega_t)$ using the formula for an arithmetic series. Note that we assume $\nabla\widetilde{f}$ exists. Next, to ensure $\omega_{t+n}$ in Equation 6 coincides with the solution $\omega(t + \epsilon)$ of the continuous modified flow $\dot{\omega} = \widetilde{f}(\omega)$ for small but finite $\epsilon$, we let the number of steps $n \to \infty$ while setting $\alpha = \epsilon/n$,

$$
\begin{align}
\omega(t + \epsilon) &= \omega(t) + \epsilon\widetilde{f}(\omega(t)) + (\epsilon^2/2)\nabla\widetilde{f}(\omega(t))\widetilde{f}(\omega(t)) + O(\epsilon^3) \tag{7} \\
&= \omega(t) + \epsilon f(\omega(t)) + \epsilon^2\left(f_1(\omega(t)) + (1/2)\nabla f(\omega(t))f(\omega(t))\right) + O(\epsilon^3). \tag{8}
\end{align}
$$

We have replaced $\widetilde{f}(\omega)$ with its definition from Equation 3. As we will see below, Equation 8 is the key component of backward error analysis, which describes the path taken when integrating the continuous modified flow $\widetilde{f}(\omega)$ with vanishing learning rates over a discrete time step of length $\epsilon$.

Notice that we have assumed that the Taylor expansion in Equation 8 converges, while the higher order terms at $O(\epsilon^3)$ will contain higher order derivatives of the original flow $f(\omega)$. Backward error analysis therefore implicitly assumes that $f(\omega)$ is an analytic function in the vicinity of the current parameters $\omega$. We refer the reader to Hairer et al. (2006) for a detailed introduction.

**Gradient descent:** As a simple example, we will now derive the first order correction $f_1(\omega)$ of the modified flow for GD. First, we recall that the discrete updates obey $\omega_{i+1} = \omega_i - \epsilon\nabla C(\omega_i)$, and we therefore fix $f(\omega) = -\nabla C(\omega)$. In order to ensure that the continuous modified flow coincides with this discrete update, we need all terms at $O(\epsilon^2)$ and above in Equation 8 to vanish. At order $\epsilon^2$, this implies that $f_1(\omega) + (1/2)\nabla\nabla C(\omega)\nabla C(\omega) = 0$, which yields the first order correction,

$$f_1(\omega) = -(1/2)\nabla\nabla C(\omega)\nabla C(\omega) = -(1/4)\nabla\left(||\nabla C(\omega)||^2\right). \tag{9}$$

We conclude that, if the learning rate $\epsilon$ is sufficiently small such that we can neglect higher order terms in Equation 3, then the discrete GD iterates lie on the path of the following ODE,

$$\dot{\omega} = -\nabla C(\omega) - (\epsilon/4)\nabla\left(||\nabla C(\omega)||^2\right) \tag{10}$$

$$= -\nabla\widetilde{C}_{GD}(\omega). \tag{11}$$

Equation 11 corresponds to gradient flow on the modified loss, $\widetilde{C}_{GD}(\omega) = C(\omega) + (\epsilon/4)||\nabla C(\omega)||^2$.

## 2.2 BACKWARD ERROR ANALYSIS AND STOCHASTIC GRADIENT DESCENT

We now derive our main result (Equation 1). As described in the introduction, we assume $N\%B = 0$, where $N$ is the training set size, $B$ is the batch size, and $\%$ denotes the modulo operation. The number of updates per epoch $m = N/B$, and the minibatch costs $\hat{C}_k(\omega) = (1/B)\sum_{j=kB+1}^{kB+B} C_j(\omega)$. SGD with constant learning rates obeys $\omega_{i+1} = \omega_i - \epsilon\nabla\hat{C}_{i\%m}(\omega_i)$. It is standard practice to shuffle the dataset once per epoch, but we omit this step here and instead perform our analysis over a single epoch. In Equation 6 we derived the influence of $n$ Euler steps on the flow $\widetilde{f}(\omega)$ with step size $\alpha$. Following a similar approach, we now derive the influence of $m$ SGD updates with learning rate $\epsilon$,

$$\omega_m = \omega_0 - \epsilon\nabla\hat{C}_0(\omega_0) - \epsilon\nabla\hat{C}_1(\omega_1) - \epsilon\nabla\hat{C}_2(\omega_2) - ... \tag{12}$$

$$= \omega_0 - \epsilon\sum_{j=0}^{m-1}\nabla\hat{C}_j(\omega_0) + \epsilon^2\sum_{j=0}^{m-1}\sum_{k<j}\nabla\nabla\hat{C}_j(\omega_0)\nabla\hat{C}_k(\omega_0) + O(m^3\epsilon^3) \tag{13}$$

$$= \omega_0 - m\epsilon\nabla C(\omega_0) + \epsilon^2\xi(\omega_0) + O(m^3\epsilon^3). \tag{14}$$

The error in Equation 14 is $O(m^3\epsilon^3)$ since there are $O(m^3)$ terms in the Taylor expansion proportional to $\epsilon^3$. Notice that a single epoch of SGD is equivalent to a single GD update with learning rate $m\epsilon$ up to first order in $\epsilon$. Remarkably, this implies that when the learning rate is sufficiently small, there is no noise in the iterates of SGD after completing one epoch. For clarity, this observation arises because we require that each training example is sampled once per epoch. However the second order correction $\xi(\omega) = \sum_{j=0}^{m-1}\sum_{k<j}\nabla\nabla\hat{C}_j(\omega)\nabla\hat{C}_k(\omega)$ does not appear in the GD update, and it is a random variable which depends on the order of the mini-batches. In order to identify the bias introduced by SGD, we will evaluate the mean correction $\mathbb{E}(\xi)$, where we take the expectation across all possible sequences of the (non-overlapping) mini-batches $\{\hat{C}_0, \hat{C}_1, ..., \hat{C}_{m-1}\}$. Note that we hold the composition of the batches fixed, averaging only over their order. We conclude that,

$$\mathbb{E}(\xi(\omega)) = \frac{1}{2}\left(\sum_{j=0}^{m-1}\sum_{k\neq j}\nabla\nabla\hat{C}_j(\omega)\nabla\hat{C}_k(\omega)\right) \tag{15}$$

$$= \frac{m^2}{2}\nabla\nabla C(\omega)\nabla C(\omega) - \frac{1}{2}\sum_{j=0}^{m-1}\nabla\nabla\hat{C}_j\nabla\hat{C}_j \tag{16}$$

$$= \frac{m^2}{4}\nabla\left(||\nabla C(\omega)||^2 - \frac{1}{m^2}\sum_{j=0}^{m-1}||\nabla\hat{C}_j(\omega)||^2\right). \tag{17}$$

For clarity, in Equation 15 we exploit the fact that every sequence of batches has a corresponding sequence in reverse order. Combining Equations 14 and 17, we conclude that after one epoch,

$$\mathbb{E}(\omega_m) = \omega_0 - m\epsilon\nabla C(\omega_0)$$
$$+ \frac{m^2\epsilon^2}{4}\nabla\left(||\nabla C(\omega_0)||^2 - (1/m^2)\sum_{j=0}^{m-1}||\nabla\hat{C}_j(\omega_0)||^2\right) + O(m^3\epsilon^3). \tag{18}$$

Having identified the expected value of the SGD iterate after one epoch $\mathbb{E}(\omega_m)$ (for small but finite learning rates), we can now use this expression to identify the corresponding modified flow. First, we set $f(\omega) = -\nabla C(\omega)$, $t = 0$, $\omega(0) = \omega_0$, and let $\epsilon \to m\epsilon$ in Equations 3 and 8 to obtain,

$$\omega(m\epsilon) = \omega_0 - m\epsilon\nabla C(\omega_0) + m^2\epsilon^2\left(f_1(\omega_0) + (1/4)\nabla||\nabla C(\omega_0)||^2\right) + O(m^3\epsilon^3). \tag{19}$$

Next, we equate Equations 18 and 19 by setting $\omega(m\epsilon) = \mathbb{E}(\omega_m)$. We immediately identify the first order correction to the modified flow $f_1(\omega) = -(1/(4m^2))\nabla\sum_{j=0}^{m-1}||\nabla\hat{C}_j(\omega)||^2$. We therefore

conclude that, after one epoch, the expected SGD iterate $\mathbb{E}(\omega_m) = \omega(m\epsilon) + O(m^3\epsilon^3)$, where $\omega(0) = \omega_0$ and $\dot\omega = -\nabla C(\omega) + m\epsilon f_1(\omega)$. Simplifying, we conclude $\dot\omega = -\nabla \widetilde{C}_{SGD}(\omega)$, where,

$$\widetilde{C}_{SGD}(\omega) \;=\; C(\omega) + (\epsilon/4m)\sum_{k=0}^{m-1}||\nabla\hat{C}_k(\omega)||^2. \tag{20}$$

Equation 20 is identical to Equation 1, and this completes the proof of our main result. We emphasize that $\widetilde{C}_{SGD}$ assumes a fixed set of minibatches $\{\hat{C}_0, \hat{C}_1, ..., \hat{C}_{m-1}\}$. We will evaluate the expected modified loss after shuffling the dataset and sampling a new set of minibatches in Section 3.

REMARKS ON THE ANALYSIS

The phrase "for small finite learning rates" has a precise meaning in our analysis. It implies $\epsilon$ is large enough that terms of $O(m^2\epsilon^2)$ may be significant, but small enough that terms of $O(m^3\epsilon^3)$ are negligible. Our analysis is unusual, because we consider the mean evolution of the SGD iterates but ignore the variance of individual training runs. Previous analyses of SGD have usually focused on the variance of the iterates in limit of vanishing learning rates (Mandt et al., 2017; Smith & Le, 2018; Jastrzębski et al., 2018). However these works assume that each minibatch is randomly sampled from the full dataset. Under this assumption, the variance in the iterates arises at $O(\epsilon)$, while the bias arises at $O(\epsilon^2)$. By contrast, in our analysis each example is sampled once per epoch, and both the variance and the bias arise at $O(\epsilon^2)$ (for simplicity, we assume $m = N/B$ is constant). We therefore anticipate that the variance will play a less important role than is commonly supposed.

Furthermore, we can construct a specific sequence of minibatches for which the variance at $O(m^2\epsilon^2)$ vanishes, such that the evolution of a specific training run will coincide exactly with gradient flow on the modified loss of Equation 1 for small finite learning rates. To achieve this, we perform two training epochs with the sequence of minibatches $(\hat{C}_0, \hat{C}_1, ..., \hat{C}_{m-1}, \hat{C}_{m-1}, ..., \hat{C}_1, \hat{C}_0)$ (i.e., the second epoch iterates through the same set of minibatches as the first but in the opposite order). If one inspects Equations 13 to 15, one will see that reversing the second epoch has the same effect as taking the mean across all possible sequences of minibatches (it replaces the $\sum_{k<j}$ by a $\sum_{k\neq j}$).

The key limitation of our analysis is that we assume $m\epsilon = N\epsilon/B$ is small, in order to neglect terms at $O(m^3\epsilon^3)$. This is an extreme approximation, since we typically expect that $N/B$ is large. Therefore, while our work identifies the first order correction to the bias arising from finite learning rates, higher order terms in the modified flow may also play an important role at practical learning rates. We note however that previous theoretical analyses have made even more extreme assumptions. For instance, most prior work studying SGD in the small learning rate limit neglects all terms at $O(\epsilon^2)$ (for an exception, see Li et al. (2017)). Furthermore, as we show in Section 3, the learning rate often scales proportional to the batch size, such that $\epsilon/B$ is constant (Goyal et al., 2017; McCandlish et al., 2018). Therefore the accuracy of our approximations does not necessarily degrade as the batch size falls, but higher order terms may play an increasingly important role as the dataset size increases. Our experimental results in Section 2.3 and Section 4 suggest that our analysis can explain most of the generalization benefit of finite learning rate SGD for Wide-ResNets trained on CIFAR-10.

We note that to achieve the highest test accuracies, practitioners usually decay the learning rate during training. Under this scheme, the modified loss would change as training proceeds. However it is widely thought that the generalization benefit of SGD arises from the use of large learning rates early in training (Smith et al., 2018; Li et al., 2019; Jastrzebski et al., 2020; Lewkowycz et al., 2020), and popular schedules hold the learning rate constant or approximately constant for several epochs.

Finally, we emphasize that our primary goal in this work is to identify the influence of finite learning rates on training. The implicit regularization term may not be beneficial in all models and datasets.

## 2.3 AN EMPIRICAL EVALUATION OF THE MODIFIED LOSS

In order to confirm that the modified loss $\widetilde{C}_{SGD}(\omega)$ can help explain why large learning rates enhance generalization, we now verify empirically that the implicit regularizer inherent in constant learning rate SGD, $C_{reg}(\omega) = (1/4m)\sum_{k=0}^{m-1}||\nabla\hat{C}_k(\omega)||^2$, can enhance the test accuracy of deep networks. To this end, we train the same model with two different (explicit) loss functions. The first loss function $C(\omega)$ represents the original loss, while the second $C_{mod}(\omega) = C(\omega) + \lambda C_{reg}(\omega)$ is obtained from the modified loss $\widetilde{C}_{SGD}(\omega)$ by replacing the learning rate $\epsilon$ with an explicit regular-

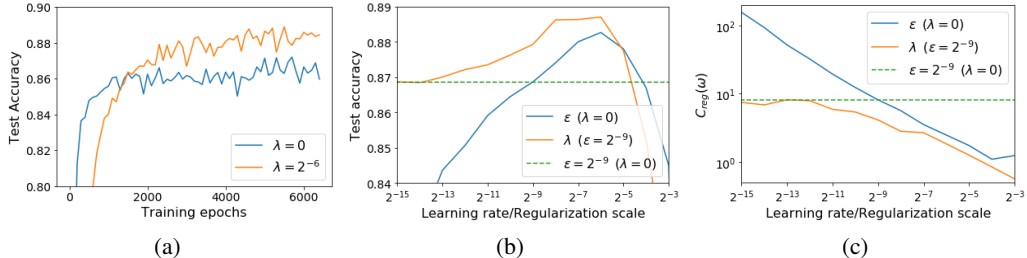

Figure 1: (a) Explicitly including the implicit regularizer in the loss improves the test accuracy when training with small learning rates. (b) The optimal regularization coefficient $\lambda_{opt} = 2^{-6}$ is equal to the optimal learning rate $\epsilon_{opt} = 2^{-6}$. (c) Increasing either the learning rate $\epsilon$ or the regularization coefficient $\lambda$ reduces the value of the implicit regularization term $C_{reg}(\omega)$ at the end of training.

ization coefficient $\lambda$. Notice that $C_{mod}(\omega) = (1/m)\sum_{k=0}^{m-1}\left(\hat{C}_k(\omega) + (\lambda/4)||\nabla\hat{C}_k(\omega)||^2\right)$, which ensures that it is straightforward to minimize the modified loss $C_{mod}(\omega)$ with minibatch gradients. Since the implicit regularization term $C_{reg}(\omega)$ is expensive to differentiate (typically 5-10x overhead), we consider a 10-1 Wide-ResNet model (Zagoruyko & Komodakis, 2016) for classification on CIFAR-10. To ensure close agreement with our theoretical analysis, we train without batch normalization using SkipInit initialization (De & Smith, 2020). We train for 6400 epochs at batch size 32 without learning rate decay using SGD without Momentum. We use standard data augmentation including crops and random flips, and we use weight decay with $L_2$ coefficient $5 \times 10^{-4}$. We emphasize that, since we train using a finite (though very large) compute budget, the final networks may not have fully converged. This is particularly relevant when training with small learning rates. Note that we provide additional experiments on Fashion-MNIST (Xiao et al., 2017) in appendix D.

In Figure 1(a), we compare two training runs, one minimizing the modified loss $C_{mod}(\omega)$ with $\lambda = 2^{-6}$, and one minimizing the original loss $C(\omega)$. For both runs we use a small constant learning rate $\epsilon = 2^{-9}$. As expected, the regularized training run achieves significantly higher test accuracies late in training. This confirms that the implicit regularizer, which arises as a consequence of using SGD with finite learning rates, can also enhance the test accuracy if it is included explicitly in the loss. In Figure 1(b), we provide the test accuracy for a range of regularization strengths $\lambda$ (orange line). We provide the mean test accuracy of the best 5 out of 7 training runs at each regularization strength, and for each run we take the highest test accuracy achieved during the entire training run. We use a fixed learning rate $\epsilon = 2^{-9}$ for all $\lambda$. For comparison, we also provide the test accuracy achieved with the original loss $C(\omega)$ for a range of learning rates $\epsilon$ (blue line). In both cases, the test accuracy rises initially, before falling for large regularization strengths or large learning rates. Furthermore, in this network the optimal regularization strength on the modified loss $\lambda_{opt} = 2^{-6}$ is equal to the optimal learning rate on the original loss $\epsilon_{opt} = 2^{-6}$. Meanwhile when $\lambda \to 0$ the performance of the modified loss approaches the performance of the original loss at $\epsilon = 2^{-9}$ (dotted green line). We provide the corresponding training accuracies in appendix C. Finally, in Figure 1(c), we provide the values of the implicit regularizer $C_{reg}(\omega)$ at the end of training. As predicted by our analysis, training with larger learning rates reduces the value of the implicit regularization term.

In Figure 2, we take the same 10-1 Wide-ResNet model and provide the mean training and test accuracies achieved at a range of learning rates for two regularization coefficients (following the experimental protocol above). In Figure 2(a), we train on the original loss $C(\omega)$ ($\lambda = 0$), while in Figure 2(b), we train on the modified loss $C_{mod}(\omega)$ with regularization coefficient $\lambda = 2^{-6}$. From Figure 2(a), when $\lambda = 0$ there is a clear generalization benefit to large learning rates, as the learning rate that maximizes test accuracy ($2^{-6}$) is 16 times larger than the learning rate that maximizes training accuracy ($2^{-10}$). However in Figure 2(b) with $\lambda = 2^{-6}$, the learning rates that maximize the test and training accuracies are equal ($2^{-8}$). This suggests that when we include the implicit regularizer explicitly in the loss, the generalization benefit of large learning rates is diminished.

## 3 Implicit Regularization and the Batch Size

In Section 2.2, we derived the modified loss by considering the expected SGD iterate after one epoch. We held the composition of the batches fixed, averaging only over the order in which the batches

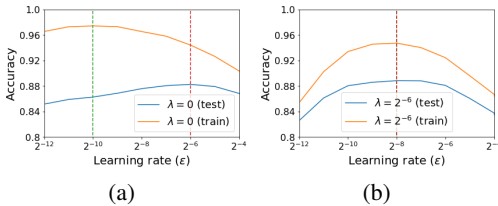

(a)                                      (b)

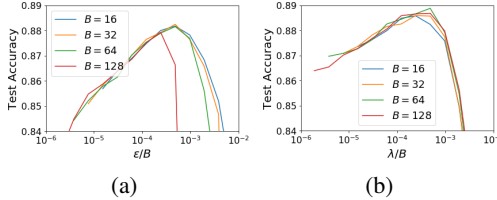

(a)                                      (b)

Figure 2: (a) There is a clear generalization benefit to large learning rates when training on the original loss $C(\omega)$ with $\lambda = 0$. (b) When we include the implicit regularizer explicitly in $C_{mod}(\omega)$ and set $\lambda = 2^{-6}$, the generalization benefit of large learning rates is diminished.

Figure 3: (a) Different batch sizes achieve the same test accuracy if the ratio of the learning rate to the batch size ($\epsilon/B$) is constant and $B$ is not too large. (b) The test accuracy is independent of the batch size if the ratio of the regularization coefficient to the batch size ($\lambda/B$) is constant.

are seen. This choice helped make clear how to explicitly include the implicit regularizer in the loss function in Section 2.3. However, in order to clarify how the implicit regularizer term depends on the batch size, we now evaluate the expected modified loss after randomly shuffling the dataset and sampling a new set of $m$ non-overlapping minibatches $\{\hat{C}_0, \hat{C}_1, ..., \hat{C}_{m-1}\}$. Since the minibatch losses $\hat{C}_i(\omega)$ are all identically distributed by symmetry, we recall Equation 2 and conclude that,

$$\mathbb{E}(\widetilde{C}_{SGD}(\omega)) = C(\omega) + (\epsilon/4)\,||\nabla C(\omega)||^2 + (\epsilon/4)\,\mathbb{E}(||\nabla \hat{C}(\omega) - \nabla C(\omega)||^2), \qquad (21)$$

where $\hat{C}(\omega)$ denotes a batch of $B$ non-overlapping examples, drawn randomly from the full dataset. To simplify equation 21, we prove in appendix A that $\mathbb{E}(||\nabla\hat{C}(\omega) - \nabla C(\omega)||^2) = \frac{(N-B)}{(N-1)}\frac{\Gamma(\omega)}{B}$, where $\Gamma(\omega) = (1/N)\sum_{i=1}^{N}||\nabla C_i(\omega) - \nabla C(\omega)||^2$. We therefore obtain,

$$\mathbb{E}(\widetilde{C}_{SGD}(\omega)) = C(\omega) + \frac{\epsilon}{4}\,||\nabla C(\omega)||^2 + \frac{(N-B)}{(N-1)}\frac{\epsilon}{4B}\,\Gamma(\omega). \qquad (22)$$

Note that $\Gamma(\omega)$ is the trace of the empirical covariance matrix of the per-example gradients. We have not assumed that the minibatch gradients are Gaussian distributed, however if the per-example gradients are heavy tailed (Simsekli et al., 2019) then $\Gamma(\omega)$ may diverge, in which case the expected value of the modified loss is ill-defined. Equation 22 shows that the implicit regularization term of SGD has two contributions. The first term is proportional to the learning rate $\epsilon$, and it penalizes the norm of the full batch gradient. The second term is proportional to the ratio of the learning rate to the batch size $\epsilon/B$ (assuming $N \gg B$), and it penalizes the trace of the covariance matrix. To interpret this result, we assume that the minibatch gradients are diverse, such that $(\Gamma(\omega)/B) \gg ||\nabla C(\omega)||^2$. This assumption guarantees that increasing the batch size reduces the error in the gradient estimate. In this limit, the second term above will dominate, and therefore different batch sizes will experience the same implicit regularization so long as the ratio of the learning rate to the batch size is constant.

To verify this claim, in Figure 2.3 we plot the mean test accuracies achieved on a 10-1 Wide-ResNet, trained on CIFAR-10 with a constant learning rate, for a range of learning rates $\epsilon$, regularization coefficients $\lambda$ and batch sizes $B$. As expected, in Figure 3(a), training on the original loss $C(\omega)$ for 6400 epochs, we see that different batch sizes achieve similar test accuracies so long as the ratio $\epsilon/B$ is constant and the batch size is not too large. We note that this linear scaling rule is well known and has been observed in prior work (Goyal et al., 2017; Smith & Le, 2018; Jastrzębski et al., 2018; Zhang et al., 2019). To confirm that this behaviour is consistent with the modified loss, in Figure 3(b) we fix the learning rate $\epsilon = 2^{-9}$ and train on $C_{mod}(\omega)$ at a range of regularization strengths $\lambda$ for 10 million steps. As expected, different batch sizes achieve similar test accuracy so long as the ratio $\lambda/B$ is constant. We note that we expect this phenomenon to break down for very large batch sizes, however we were not able to run experiments in this limit due to computational constraints.

For very large batch sizes, the first implicit regularization term in Equation 22 dominates, the linear scaling rule breaks down, and the bias of SGD is similar to the bias of GD identified by Barrett & Dherin (2021). We expect the optimal learning rate to be independent of the batch size in this limit, as observed by McCandlish et al. (2018) and Smith et al. (2020). Convergence bounds also predict a transition between a small batch regime where the optimal learning rate $\epsilon \propto B$ and a large batch regime where the optimal learning rate is constant (Ma et al., 2018; Zhang et al., 2019). However these analyses identify the learning rate which minimizes the training loss. Our analysis compliments these claims by explaining why similar conclusions hold when maximizing test accuracy.

# 4 FINITE LEARNING RATES AND STOCHASTIC DIFFERENTIAL EQUATIONS

In the previous two sections, we argued that the use of finite learning rates and small batch sizes introduces implicit regularization, which can enhance the test accuracy of deep networks. We analyzed this effect using backward error analysis (Hairer et al., 2006; Li et al., 2017; Barrett & Dherin, 2021), but many previous papers have argued that this effect can be understood by interpreting small batch SGD as the discretization of an SDE (Mandt et al., 2017; Smith & Le, 2018; Jastrzębski et al., 2018; Park et al., 2019). In this section, we compare this popular perspective with our main results from Sections 2 and 3. To briefly recap, in the SDE analogy a single gradient update is given by $\omega_{i+1} = \omega_i - \epsilon \nabla \hat{C}(\omega_i)$, where $\hat{C}$ denotes a random batch of $B$ non-overlapping training examples. Notice that in the SDE analogy, since examples are drawn randomly from the full dataset, there is no guarantee that each training example is sampled once per epoch. Assuming $N \gg B \gg 1$ and that the gradients are not heavy tailed, the central limit theorem is applied to model the noise in an update by a Gaussian noise source $\xi$ whose covariance is inversely proportional to the batch size:

$$\omega_{i+1} = \omega_i - \epsilon \big( \nabla C(\omega_i) + \xi_i / \sqrt{B} \big) \quad = \quad \omega_i - \epsilon \nabla C(\omega_i) + \sqrt{\epsilon T} \xi_i. \tag{23}$$

This assumes $\mathbb{E}(\xi_i) = 0$ and $\mathbb{E}(\xi_i \xi_j^T) = F(\omega) \delta_{ij}$, where $F(\omega)$ is the covariance matrix of the per example gradients, and we define the "temperature" $T = \epsilon / B$. The SDE analogy notes that Equation 23 is identical to the Euler discretization of an SDE with step size $\epsilon$ and temperature $T$ (Gardiner et al., 1985). Therefore one might expect the SGD iterates to remain close to this underlying SDE in the limit of small learning rates ($\epsilon \to 0$). In this limit, the temperature defines the influence of mini-batching on the dynamics, and it is therefore often assumed that the temperature also governs the generalization benefit of SGD (Smith & Le, 2018; Jastrzębski et al., 2018; Park et al., 2019).

However this conclusion from the SDE analogy is inconsistent with our analysis in Section 2. To see this, note that in Section 2 we assumed that each training example is sampled once per epoch, as recommended by practitioners (Bottou, 2012), and showed that under this assumption there is *no noise* in the dynamics of SGD up to first order in $\epsilon$ after one epoch of training. The SDE analogy therefore relies on the assumption that minibatches are sampled randomly from the full dataset. Furthermore, SGD only converges to the underlying SDE when the learning rate $\epsilon \to 0$, but in this limit the temperature $T \to 0$ and SGD converges to gradient flow (Yaida, 2019). We must use a finite learning rate to preserve a finite temperature, but at any finite learning rate the distributions of the SGD iterates and the underlying SDE may differ. We now provide intriguing empirical evidence to support our contention that the generalization benefit of SGD arises from finite learning rates, not the temperature of an associated stochastic process. First, we introduce a modified SGD update rule:

**n-step SGD:** *Apply $n$ gradient descent updates sequentially on the same minibatch with bare learning rate $\alpha$, effective learning rate $\epsilon = n\alpha$ and batch size $B$. Sample the next minibatch and repeat.*

To analyze n-step SGD, we consider the combined influence of $n$ updates on the same minibatch:

$$\omega_{i+1} \quad = \quad \omega_i - \alpha \nabla \hat{C}(\omega_i) - \alpha \nabla \hat{C}(\omega_i - \alpha \nabla \hat{C}(\omega_i)) + ... \tag{24}$$

$$= \quad \omega_i - n\alpha \nabla \hat{C}(\omega_i) + O(n^2 \alpha^2) \tag{25}$$

$$= \quad \omega_i - \epsilon \nabla C(\omega_i) + \sqrt{\epsilon T} \xi_i + O(\epsilon^2). \tag{26}$$

Equations 23 and 26 are identical up to first order in $\epsilon$ but they differ at $O(\epsilon^2)$ and above. Therefore, if minibatches are randomly sampled from the full dataset, then the dynamics of standard SGD and n-step SGD should remain close to the same underlying SDE in the limit $\epsilon \to 0$, but their dynamics will differ when the learning rate is finite. We conclude that if the dynamics of SGD is close to the continuum limit of the associated SDE, then standard SGD and n-step SGD ought to achieve similar test accuracies after the same number of training epochs. However if, as we argued in Section 2, the generalization benefit of SGD arises from finite learning rate corrections at $O(\epsilon^2)$ and above, then we should expect the performance of standard SGD and n-step SGD to differ. For completeness, we provide a backward error analysis of n-step SGD in appendix B. In line with Section 2 (and best practice), we assume each training example is sampled once per epoch. We find that after one epoch, the expected n-step SGD iterate $\mathbb{E}(\omega_m) = \omega(m\epsilon) + O(m^3 \epsilon^3)$, where $\omega(0) = \omega_0$, $\dot{\omega} = -\nabla \widetilde{C}_{nSGD}(\omega)$ and $\widetilde{C}_{nSGD}(\omega) = C(\omega) + (\epsilon/4mn) \sum_{i=0}^{m-1} ||\nabla C_i(\omega)||^2$. The scale of the implicit regularizer is proportional to $\alpha = \epsilon/n$, which implies that the implicit regularization is suppressed as $n$ increases if we hold $\epsilon$ constant. As expected, we recover Equation 1 when $n = 1$.

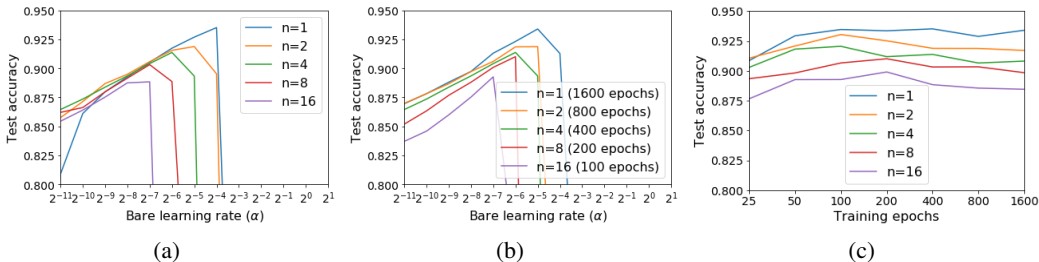

Figure 4: (a) When training for 400 epochs, smaller values of $n$ are stable at larger bare learning rates $\alpha$, and this enables them to achieve higher test accuracies. (b) Similar conclusions hold when training for a fixed number of updates. (c) We show the test accuracy at the optimal learning rate for a range of epoch budgets. We find that smaller values of $n$ consistently achieve higher test accuracy.

In Figure 4(a), we plot the performance of n-step SGD at a range of bare learning rates $\alpha$, when training a 16-4 Wide-ResNet on CIFAR-10 for 400 epochs using SkipInit (De & Smith, 2020) at batch size 32. Each example is sampled once per epoch. We introduce a learning rate decay schedule, whereby we hold the learning rate constant for the first half of training, before decaying the learning rate by a factor of 2 every remaining tenth of training, and we provide the mean test accuracy of the best 5 out of 7 training runs for each value of $\alpha$. The optimal test accuracy drops from $93.5\%$ when $n = 1$ (standard SGD) to $88.8\%$ when $n = 16$. This occurs even though all values of $n$ perform the same number of training epochs, indicating that 16-step SGD performed 16 times more gradient updates. These results suggest that, at least for this model and dataset, the generalization benefit of SGD is not controlled by the temperature of the associated SDE, but instead arises from the implicit regularization associated with finite learning rates. When we increase $n$ we reduce the largest stable bare learning rate $\alpha$, and this suppresses the implicit regularization benefit, which reduces the test accuracy. We also verify in Figure 4(b) that similar conclusions arise if we hold the number of parameter updates fixed (such that the number of training epochs is inversely proportional to $n$). Smaller values of $n$ are stable at larger bare learning rates and achieve higher test accuracies. Finally we confirm in Figure 4(c) that the test accuracy degrades as $n$ increases even if one tunes both the learning rate and the epoch budget independently for each value of $n$, thus demonstrating that n-step SGD consistently achieves lower test accuracies as n increases. Note that we provide additional experiments on Fashion-MNIST (Xiao et al., 2017) in appendix D.

## 5 DISCUSSION

Many authors have observed that large learning rates (Li et al., 2019; Lewkowycz et al., 2020), and small batch sizes (Keskar et al., 2017; Smith et al., 2020), can enhance generalization. Most theoretical work has sought to explain this by observing that increasing the learning rate, or reducing the batch size, increases the variance of the SGD iterates (Smith & Le, 2018; Jastrzębski et al., 2018; Chaudhari & Soatto, 2018). We take a different approach, and note that when the learning rate is finite, the SGD iterates are also biased (Roberts, 2018). Backward error analysis (Hairer et al., 2006; Li et al., 2017; Barrett & Dherin, 2021) provides a powerful tool that computes how this bias accumulates over multiple parameter updates. Although this work focused on GD and SGD, we anticipate that backward error analysis could also be used to clarify the role of finite learning rates in adaptive optimizers like Adam (Kingma & Ba, 2015) or Natural Gradient Descent (Amari, 1998).

We note however that backward error analysis assumes that the learning rate is small (though finite). It therefore does not capture the chaotic or oscillatory dynamics which arise when the learning rate is close to instability. At these very large learning rates the modified loss, which is defined as a Taylor series in powers of the learning rate, does not converge. Lewkowycz et al. (2020) recently argued that the test accuracies of wide networks trained with full batch gradient descent on quadratic losses are maximized for large learning rates close to divergence. In this "catapult" regime, the GD iterates oscillate along high curvature directions and the loss may increase early in training. It remains an open question to establish whether backward error analysis fully describes the generalization benefit of small batch SGD, or if these chaotic or oscillatory effects also play a role in some networks.

ACKNOWLEDGMENTS

We would like to thank Jascha Sohl-Dickstein, Razvan Pascanu, Alex Botev, Yee Whye Teh and the anonymous reviewers for helpful discussions and feedback on earlier versions of this manuscript.

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

## A  THE EXPECTED NORM OF A MINIBATCH GRADIENT

To keep the notation clean, we define $X_i = (\nabla C_i(\omega) - \nabla C(\omega))$. We also recall for clarity that the expectation value $\mathbb{E}(...)$ is taken over all possible random shuffles of the indices $i$. Therefore,

$$
\begin{aligned}
\mathbb{E}(||(\nabla \hat{C}(\omega) - \nabla C(\omega))||^2) &= \frac{1}{B^2}\mathbb{E}\left(\sum_{i=1}^{B}\sum_{j=1}^{B} X_i \cdot X_j\right) && (27) \\
&= \frac{B}{B^2}\mathbb{E}(X_i \cdot X_i) + \frac{B(B-1)}{B^2}\mathbb{E}(X_i \cdot X_{j\neq i}) && (28) \\
&= \frac{1}{NB}\sum_{i=1}^{N} X_i \cdot X_i + \frac{(B-1)}{B}\frac{1}{N(N-1)}\sum_{i=1}^{N}\sum_{j\neq i} X_i \cdot X_j && (29) \\
&= \frac{1}{NB}\sum_{i=1}^{N} X_i \cdot X_i + \frac{(B-1)}{BN(N-1)}\sum_{i=1}^{N}\sum_{j=1}^{N} (X_i \cdot X_j(1 - \delta_{ij})). &&
\end{aligned}
$$

Note that we obtain Equation 28 by counting the number of diagonal and off-diagonal terms in the sum in Equation 27. Next, we recall that $\sum_{i=1}^{N} X_i = \sum_{i=1}^{N}(\nabla C_i(\omega) - \nabla C(\omega)) = 0$. Therefore,

$$
\begin{aligned}
\mathbb{E}(||(\nabla \hat{C}(\omega) - \nabla C(\omega))||^2) &= \frac{1}{NB}\sum_{i=1}^{N} X_i \cdot X_i - \frac{(B-1)}{BN(N-1)}\sum_{i=1}^{N} X_i \cdot X_i && (30) \\
&= \frac{1}{NB}\left(1 - \frac{(B-1)}{(N-1)}\right)\sum_{i=1}^{N} X_i \cdot X_i && (31) \\
&= \frac{(N-B)}{(N-1)}\frac{\Gamma(\omega)}{B}, && (32)
\end{aligned}
$$

where $\Gamma(\omega) = (1/N)\sum_{i=1}^{N} X_i \cdot X_i = (1/N)\sum_{i=1}^{N} ||\nabla C_i(\omega) - \nabla C(\omega)||^2$. We can immediately identify $\Gamma(\omega)$ as the trace of the empirical covariance matrix of the per-example gradients.

## B  A BACKWARD ERROR ANALYSIS FOR n-STEP SGD

Under n-step SGD, we apply $n$ gradient descent updates on the same minibatch with bare learning rate $\alpha$ and batch size $B$. After $n$ updates, we sample the next minibatch and repeat. For convenience, we define the effective learning rate $\epsilon = n\alpha$. After one minibatch ($n$ parameter updates),

$$
\begin{aligned}
\omega_{i+1} &= \omega_i - \alpha\nabla\hat{C}_i(\omega_i) - \alpha\nabla\hat{C}_i(\omega_i - \alpha\nabla\hat{C}_i(\omega_i)) + ... && (33) \\
&= \omega_i - n\alpha\nabla\hat{C}_i(\omega_i) + (n/2)(n-1)\alpha^2\nabla\nabla\hat{C}_i(\omega_i)\nabla\hat{C}_i(\omega_i) + O(n^3\alpha^3) && (34) \\
&= \omega_i - \epsilon\nabla\hat{C}_i(\omega_i) + (1/4)(1 - 1/n)\epsilon^2\nabla\left(||\nabla\hat{C}_i(\omega_i)||^2\right) + O(\epsilon^3). && (35)
\end{aligned}
$$

After one epoch (including terms up to second order in $\epsilon$),

$$
\begin{aligned}
\omega_m = \omega_0 &- \epsilon\nabla\hat{C}_0(\omega_0) + (1/4)(1 - 1/n)\epsilon^2\nabla\left(||\nabla\hat{C}_0(\omega_0)||^2\right) \\
&- \epsilon\nabla\hat{C}_1(\omega_1) + (1/4)(1 - 1/n)\epsilon^2\nabla\left(||\nabla\hat{C}_1(\omega_1)||^2\right) \\
&- ... \\
&- \epsilon\nabla\hat{C}_{m-1}(\omega_{m-1}) + (1/4)(1 - 1/n)\epsilon^2\nabla\left(||\nabla\hat{C}_{m-1}(\omega_{m-1})||^2\right) + O(\epsilon^3). \quad (36)
\end{aligned}
$$

To simplify this expression, we note that $\omega_{i+1} = \omega_i - \epsilon\nabla\hat{C}_i(\omega_i) + O(\epsilon^2)$. We can therefore re-use our earlier analysis from Section 2.2 of the main text (see Equation 13 for comparison) to obtain,

$$
\begin{aligned}
\omega_m &= \omega_0 - m\epsilon\nabla C(\omega_0) + \epsilon^2\sum_{j=0}^{m-1}\sum_{k<j}\nabla\nabla\hat{C}_j(\omega_0)\nabla\hat{C}_k(\omega_0) \\
&\quad + (1/4)(1 - 1/n)\epsilon^2\sum_{i=0}^{m-1}\nabla\left(||\nabla\hat{C}_i(\omega_0)||^2\right) + O(\epsilon^3). && (37)
\end{aligned}
$$

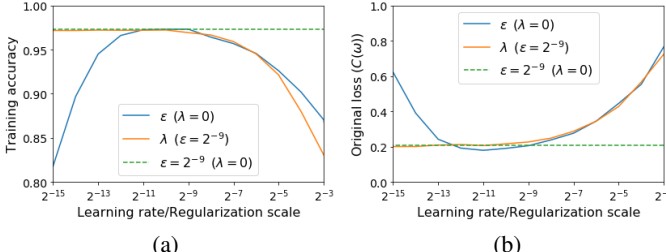

(a)     (b)

Figure 5: (a) When training on the original loss $C(\omega)$, the training accuracy achieved within 6400 epochs is maximized when the learning rate $\epsilon = 2^{-9}$. Smaller learning rates have not yet converged, while larger learning rates reach a plateau. When training on the modified loss $C_{mod}(\omega)$ with fixed learning rate $\epsilon = 2^{-9}$ and regularization coefficient $\lambda$, we achieve high training accuracies when $\lambda$ is small, but are unable to achieve high training accuracies when $\lambda$ is large. (b) We plot the value of the original loss $C(\omega)$ at the end of training. Remarkably, the training losses obtained when training on the original loss with a large learning rate are similar to the training losses achieved when training on the modified loss with small learning rate ($\epsilon = 2^{-9}$) and a large regularization coefficient $\lambda$.

Taking the expectation over all possible batch orderings (see Equations 15 to 18), we obtain,

$$\mathbb{E}(\omega_m) = \omega_0 - m\epsilon\nabla C(\omega_0) + \frac{m^2\epsilon^2}{4}\nabla\left(||\nabla C(\omega_0)||^2 - \frac{1}{m^2n}\sum_{i=0}^{m-1}||\nabla \hat{C}_i(\omega_0)||^2\right) + O(m^3\epsilon^3). \quad (38)$$

Fixing $f(\omega) = -\nabla C(\omega)$ and equating Equation 38 with the continuous modified flow in Equation 19 by setting $\mathbb{E}(\omega_m) = \omega(m\epsilon)$, we identify the modified flow $\dot{\omega} = -\nabla\widetilde{C}_{nSGD} + O(m^2\epsilon^2)$, where,

$$\widetilde{C}_{nSGD}(\omega) = C(\omega) + \frac{\epsilon}{4mn}\sum_{i=0}^{m-1}||\nabla\hat{C}_i(\omega)||^2. \quad (39)$$

Comparing Equation 39 to Equation 1, we note that the modified losses of SGD and n-step SGD coincide when $n = 1$. However for n-step SGD when $n > 1$, the strength of the implicit regularization term is proportional to the scale of the bare learning rate $\alpha = \epsilon/n$, not the effective learning rate $\epsilon$.

## C   TRAINING LOSSES

In Figure 1(b) of Section 2.3 in the main text, we compared the test accuracies achieved when training on the original loss $C(\omega)$ at a range of learning rates $\epsilon$, to the test accuracies achieved when training on the modified loss $C_{mod}(\omega)$ at fixed learning rate $\epsilon = 2^{-9}$ and a range of regularization coefficients $\lambda$. For completeness, in Figure 5, we provide the corresponding training accuracies, as well as the final values of the original loss $C(\omega)$. Remarkably, large learning rates and large regularization coefficients achieve similar training accuracies and similar original losses. This suggests that the implicit regularization term in the modified loss of SGD ($\widetilde{C}_{SGD}(\omega)$) may help explain why the training accuracies and losses often exhibit plateaus when training with large learning rates.

## D   ADDITIONAL RESULTS ON FASHION-MNIST

In this section we provide additional experiments on the Fashion-MNIST dataset (Xiao et al., 2017), which comprises 10 classes, 60000 training examples and 10000 examples in the test set. We consider a simple fully connected MLP which comprises 3 nonlinear layers, each with width 4096 and ReLU activations, and a final linear softmax layer. We apply a simple data pipeline which first applies per-image standardization and then flattens the input to a 784 dimensional vector. We do not apply data augmentation and we train using vanilla SGD without learning rate decay for all experiments. We perform seven training runs for each combination of hyper-parameters and show the mean performance of the best five (to ensure our results are not skewed by a single failed run). We use a batch size $B = 16$ unless otherwise specified, and we do not use weight decay.

We note that this model is highly over-parameterized. Unlike the Wide-ResNet we considered in the main text we consistently achieve 100% training accuracy if the learning rate is not too large.

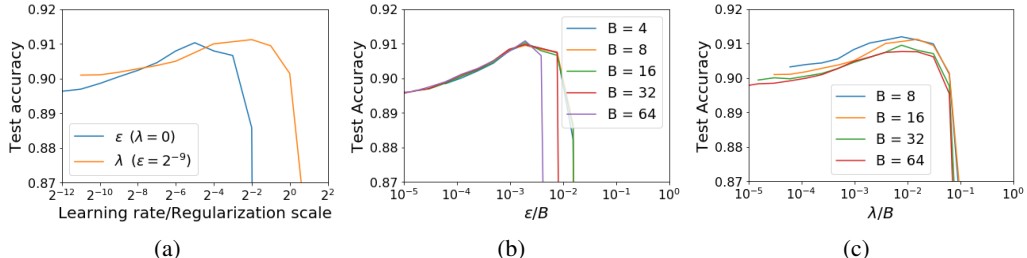

Figure 6: (a) Tuning the regularization scale $\lambda$ has a similar influence on test accuracy to tuning the learning rate $\epsilon$, although unlike our CIFAR-10/Wide-ResNet experiments the optimal values of $\lambda$ and $\epsilon$ do not coincide. (b) We obtain similar accuracies at different batch sizes if the ratio $\epsilon/B$ is constant. (c) We obtain similar accuracies at different batch sizes if the ratio $(\lambda/B)$ is constant.

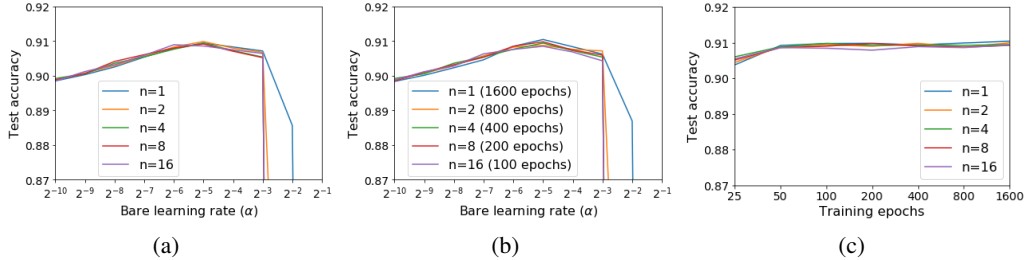

Figure 7: (a) We train with n-step SGD for 400 epochs, and achieve similar test accuracies for different values of n, so long as the bare learning rate $\alpha$ is the same. This is consistent with backward error analysis of n-step SGD, but contradicts the predictions of the SDE analogy. (b) We observe a similar phenomenon when training for a fixed number of parameter updates. (c) Different values of n achieve similar test accuracies at their optimal learning rate, across a range of epoch budgets.

In Figure 6(a), we train for 400 epochs, and we compare the effect of tuning the learning rate $\epsilon$ when training on the original loss, to the effect of tuning the explicit regularization strength $\lambda$ (with $\epsilon = 2^{-9}$). As observed in the main text, tuning the explicit regularizer has a similar effect on the test accuracy to tuning the learning rate. Surprisingly, the optimal values of $\epsilon$ and $\lambda$ differ by a factor of 8. However we note that the optimal learning rate is $\epsilon = 2^{-5}$, while the explicit regularizer already achieves a very similar test accuracy at $\lambda = 2^{-4}$ (just a factor of two larger), before reaching a higher maximum test accuracy at $\lambda = 2^{-2}$. In Figure 6(b), we train for 400 epochs on the original loss and compare the test accuracies achieved for a range of batch sizes at different learning rates. As observed in the main text, the test accuracy is determined by the ratio of the learning rate to the batch size. Meanwhile in Figure 6(c), we plot the test accuracy achieved after training for 1.5 million steps on the modified loss with learning rate $\epsilon = 2^{-9}$ and regularization coefficient $\lambda$. Once again, we find that the test accuracy achieved is primarily determined by the ratio of the regularization coefficient to the batch size, although smaller batch sizes also achieve slightly higher accuracies.

Finally, in Figure 7 we train using n-step SGD (see Section 4) on the original loss at a range of bare learning rates $\alpha$. In Figure 7(a) we train for 400 epochs, while in Figure 7(b) we train for 6 million updates. We recall that the SDE analogy predicts that the generalization benefit of n-SGD would be determined by the effective learning rate $\epsilon = n\alpha$. By contrast, backward error analysis predicts that the generalization benefit for small learning rates would be controlled by the bare learning rate $\alpha$, but that higher order terms may be larger for larger values of $n$. We find that the test accuracy in both figures is governed by the bare learning rate $\alpha$, not the effective learning rate $\epsilon = n\alpha$, and therefore these results are inconsistent with the predictions from the SDE analysis in prior work.

Note that Figure 7 has a surprising implication. It suggests that, for this model, while there is a largest stable bare learning rate we cannot exceed, we can repeatedly apply updates obtained on the same batch of training examples without suffering a significant degradation in test accuracy. We speculate that this may indicate that the gradients of different examples in this over-parameterized model are close to orthogonal (Sankararaman et al., 2020).

