# OpenReview forum: "On the Origin of Implicit Regularization in Stochastic Gradient Descent"
_ICLR.cc/2021/Conference — ICLR 2021 Poster_

### Official Review · AnonReviewer1 · 2020-10-27
**Missing a discussion of the scope of the results and the assumptions that go into it**

**Rating:** 7
**Confidence:** 4

**Review:**

## Summary

Using backward error analysis, the paper argues that SGD with small but finite step sizes stays on the path of a gradient flow ODE of a modified loss, which penalizes the squared norms of the mini-batch gradients. This offers a possible explanation of the empirically observed positive effect of (relatively) large step sizes on generalization performance. The paper further contests previous findings based on a vanishing step size assumption.


## Rating

Similar to several recent works, this paper tries to explain certain aspects of stochastic gradient descent using a continuous time approximation. In contrast to existing works, it explicitly accounts for the effect of finite step sizes, which I think is a very interesting direction and surfaces several interesting aspects. I also welcome and endorse the critical discussion of prior work based on infinitesimal step size assumptions. Overall, the paper was interesting and pleasant to read. To the very best of my knowledge, all mathematical derivations are technically correct.

However—as the authors themselves note in their critique of SDE approximations to SGD—the devil is in the details with continuous time approximations. In my opinion, that makes is absolutely crucial to discuss the scope of the results carefully and transparently, including a critical discussion on assumptions made and simplifications that go into the continuous-time model. In my opinion, this paper fails to deliver that, which is why I recommend rejection. Below, I am asking for clarification on various points and would encourage the authors to respond to the major points in the rebuttal phase.


## Major Comments


1) The main result says that the *expected* SGD iterate after a *single* epoch lands close to the path of a gradient flow ODE on a modified loss. Unless I am missing something, this fundamentally fails to capture the behavior over multiple epochs. The analysis only guarantees that, from any given starting point $\omega_0$, the expected iterate after one epoch of SGD ends up close to the ODE path starting from $\omega_0$. Unless I am missing something, this does *not* imply that two epochs of SGD starting from $\omega_0$ end up on that path. We can not simply chain two epochs together: The first epoch only stays on the path in expectation, but any realization of that random variable will deviate from the path, which affects the initial condition of the next epoch. Intuitively, one needs to get a handle on the variance of the iterate as well in order to give guarantees for multiple epochs. Is this understanding correct? If so, to what extent can insights about a single epoch of SGD be transferred to practical settings?

2) Comment (1) hints at a larger (but vague) point that the paper is trying to characterize a *stochastic* optimization procedure with a solution of a *deterministic* gradient flow ODE. It does so by focusing on the *expectation* of the iterate, which might be an approach to highlight certain aspects, but it will never give a full picture. Why wouldn’t we also be interested in the covariance of the iterates? The limitations of this characterization should be discussed thoroughly in the paper.

3) In Section 2, the composition of the minibatches is assumed to be fixed and the randomness only comes from their ordering. The paper says: "It is standard practice to shuffle the dataset once per epoch, but this step does not affect our analysis and we omit it for brevity.“ I don’t think that statement is justified with respect to the result in Eq. (1), given that the modified loss depends on the minibatch composition. Therefore, would we reshuffle the dataset after each epoch, the modified loss would change from one epoch to the next. Later, in Section 3, the expectation is additionally taken over the composition of the batches. Why is the result presented in these two distinct steps? None of the key findings of the paper seems to rely on the intermediate fixed-composition result. It also doesn’t reflect the common practice of reshuffling the entire dataset and then traversing it, which simultaneously randomizes the composition and ordering of batches. So why not give the result of Eq. (22) directly? It is also the more intuitive result, invoking the trace of the gradient covariance matrix, which also appears in prior work on continuous time approximations of SGD.

4) While the analysis tries to account for finite step sizes, it still seems to assume step sizes that are orders of magnitude smaller than those used in practice. In particular, when going from Eq. (12) to Eq. (13), each minibatch cost function is equated with its second-order Taylor approximation around the starting point $\omega_0$. This is a *drastic* approximation and I don’t see any justification for why this should be anywhere near accurate for practical settings. For large datasets and moderate batch sizes, the number of updates in one epoch will be in the thousands. For realistic step size choices, a second-order Taylor expansion around the starting point will probably be rather poor after a handful of SGD updates, no?

5) The paper strongly emphasizes the assumption of sampling data points without replacement. While sampling without replacement is indeed the usual setting in practice, most of the stochastic optimisation literature builds on the assumption of sampling with replacement. And to my knowledge, no major differences (in terms of generalization performance) have been reported in the literature between the two approaches.
    a) Can the analysis presented in the paper be extended to setting of sampling with replacement? It seems to me that this should be straight-forward. Equations (12) and (13) should hold also when each minibatch is obtained from sampling with replacement. In that case, the expectation of the second-order correction term should directly give a result akin to Eq. (22). If that is in fact possible, it should definitely be added to the paper.
    b) If that is not possible, what prevents the application and is this a technicality or would you actually expect substantially different behavior in terms of generalization?
    c) It would also have been nice to see the experiments repeated with sampling with replacement to check empirically whether the findings hold in that case?

6) Something that bugs me from an optimization perspective is that the smoothness properties of the problem do not enter this analysis at all. For example, you write (near the bottom of page 4) that “our analysis assumes $m\epsilon = N\epsilon / B$ is small.” However, any given loss function $C(w)$ can be rescaled by a constant $M\gg 1$ while scaling the step size with $1/M$. This leaves the behavior of SGD unaffected while making the step size arbitrarily small. Why does that not enter into the analysis? It probably relates to my comment (4), seeing that the step sizes are assumed to be so small that they are not restricted by the smoothness of the function.


## Minor Comments

7) The paper derives the implicit regularizer and provides empirical evidence that it can partially explain the benefits of large step sizes for generalization. However, very little attention is given to the regularization term itself and to the question *why* this regularizer might be beneficial. The only comment speaking to that is that the regularizer penalizes “sharp” regions. I would like to see this discussion expanded and connected to the recent literature.

8) At the end of page 6, you write about the large batch size regime and say that the “we expect the optimal learning rate to be independent of the batch size in this limit.” It would have been great to substantiate that conjecture with an experiment and/or to refer to specific experiments done in prior work.

9) You repeatedly use the phrase “small but finite learning rates”. If my understanding is correct, that has phrase has a very precise meaning in the context of this work, namely that terms of order $O(\epsilon^3)$ are vanishingly small while terms that a quadratic or linear in $\epsilon$ can not be ignored. (This is in contrast to prior work that also ignores quadratic terms.) Maybe this could be stated clearly the first time you use this phrase.


## Typos / Style

- I think you should capitalize references to sections, equations, figures, et cetera.
- The bib file could really need some love. You are citing the arXiv versions for several papers that have been published in peer-reviewed venues. Capitalization in paper titles is messed up (e.g., “sgd”).

## Edit after Rebuttal

I thank the authors for their engagement with my review. Many of my comments and questions have been resolved and, consequently, I have increase my score and **recommend accepting this paper.**

---

> ### Author Response · Authors · 2020-11-16
> **Response part 1**
>
> We would like to thank you for your review, and for your constructive feedback which we believe will help us to significantly improve the paper. We address all of your comments below, and we are currently working to incorporate the changes you suggest in the text.
>
> For readability, the numbering on our responses below match the numbering of the comments raised in your review:
>
> 1) Our analysis considers the expected value of the SGD iterates, but neglects the variance of individual training runs. We agree that this point should have been clearer in the original submission, and we will update the text to clarify.
>
>  We note that most prior work studying SGD in the limit of small learning rates has focused on the variance of the iterates, rather than the bias. This is because, under a random sampling strategy, the variance arises at $O(\epsilon)$, while the bias arises at $O(\epsilon^2)$. However under the random shuffling strategy we consider, both the variance and the bias arise only at $O(\epsilon^2)$. We therefore anticipate that the variance will play a less important role than is commonly supposed, which is why we choose to focus on the bias of the expected iterate in this work. Our experiments suggest that the bias in the iterates up to $O(\epsilon^2)$ can explain most of the generalization benefit of finite learning rate SGD, at least for Wide-ResNets/CIFAR-10. However we do agree that studying the variance of the iterates is an interesting avenue for future work, and we will update the text to clarify this point.
>
>  Intriguingly, there is a specific sequence of minibatches which leaves the bias at $O(\epsilon^2)$ unchanged, but for which the SGD iterates only exhibit variance at $O(\epsilon^3)$ and above. To achieve this we shuffle the dataset, perform a single epoch, then reverse the dataset and perform a second epoch, iterating through the same minibatches but in the opposite order. We then shuffle again and repeat. If you inspect equations 13-15 in our analysis, you will see that reversing the sequence of the minibatches every second epoch has the same effect as taking the expectation over all possible sequences (it replaces the sum over $k < j$ by a sum over $k \neq j$). Under this "reverse epoch" strategy the path taken by the modified flow will coincide exactly with the discrete iterates of SGD (for a specific training run, at the end of every second epoch).
>
> 2) See point 1.
>
> 3) We agree that the line "It is standard practice to shuffle the dataset once per epoch, but this step does not affect our analysis and we omit it for brevity“ was misleading and we will remove it from the updated text.
>
>  We also agree that equation 22 is more intuitive than equation 1, since it does not refer to specific minibatches and clarifies the role of batch size. We debated before submission whether to derive equation 22 directly or to provide both steps as in the manuscript, however we chose eventually to provide both steps. This is because equation 1 can be expressed directly as a sum over minibatch losses, and it is therefore clearer how to implement the modified loss from equation 1 in our experiments. We also believe that the derivation is easier to follow in two stages. We will add a note to clarify this in the paper.
>
> 4) Our analysis assumes that $m\epsilon = N\epsilon/B$ is small, in order to neglect terms at $O(m^3\epsilon^3)$. We agree that this is at first sight an extreme approximation and that, as we already note in the text, higher order terms are also likely to play a role in practice. We anticipate that these terms could be an interesting avenue for future work, and we will make this point clearer in the updated text. However, we believe the approximation is reasonable for the following reasons:
>
>   - Our work identifies the lowest order correction to gradient flow for SGD with finite learning rates. We believe that this is in itself a significant theoretical contribution, especially since this lowest order correction has a natural interpretation as an implicit regularizer.
>
>   - Our experiments suggest that, at least for Wide-ResNets/CIFAR-10, our analysis is sufficient to describe most of the generalization benefit of large learning rates.
>
>   - Furthermore, as we show in section 3, the optimal learning rate is usually proportional to the batch size for small batch sizes, and therefore N\epsilon/B does not grow as the batch size falls. Consequently, our approximation may be surprisingly accurate in practice, even when the number of updates in an epoch is large.
>
>   - Many previous well-cited works in this area make even more extreme approximations, yet these works have still yielded useful insights. For instance, the SDE analogy neglects all terms of $O(\epsilon^2)$ and above, while our analysis only neglects terms at $O(\epsilon^3)$.

---

> > ### Author Response · Authors · 2020-11-16
> > **Response part 2**
> >
> > 5) One can apply backwards analysis to consider SGD under random sampling (with replacement), and indeed this analysis was already presented in [1]. We will clarify this point in the text. However in this case, SGD is described by a (modified) stochastic differential equation which exhibits variance at $O(\epsilon)$. This stochastic differential equation is significantly more challenging to analyse than the ODE presented in our work, since the variance in the iterates at $O(\epsilon)$ may also give rise to a bias in the iterates at $O(\epsilon^2)$. This bias is difficult to identify in closed form, and therefore the implicit regularization effect of SGD is harder to uncover for this case. By instead requiring that each minibatch is sampled once per epoch, we eliminate the variance at $O(\epsilon)$, and consequently the bias in the iterates at $O(\epsilon^2)$ becomes explicit.
> >
> >  We will repeat our Wide-ResNet experiments sampling with replacement, and add these to the appendices (unfortunately these experiments may not be complete before the end of the rebuttal period).
> >
> > 6) The curvature of the loss does play a role in the analysis, and we will update the text to clarify this point. Note that the implicit regularization term arises from a Hessian-vector product (see equation 9), and therefore the implicit regularization itself arises from the curvature of the loss. Furthermore, our analysis assumes that $m\epsilon = N\epsilon/B$ is small, which allows us to neglect terms at $O(\epsilon^3)$ and above. However the neglected terms at $O(\epsilon^3)$ are composed of first, second and third derivatives of the minibatch losses, and the accuracy of this approximation will depend on the scales of these derivatives.
> >
> > 7) Our main goal in this work is simply to derive the lowest order corrections to gradient flow, which arise from finite learning rates. Remarkably, these corrections have a natural interpretation as an implicit regularizer, but we are not aware of any principled reason why this implicit regularizer is beneficial in practice.
> >
> >  Intuitively, the implicit regularizer is composed of two components, one of which penalizes the euclidean norm of the full batch gradient, and which might therefore be interpreted as penalizing "sharp minima". The second component, which dominates in the small batch limit, penalizes the trace of the covariance matrix of the per example gradients, and it might therefore be interpreted as favouring consistent solutions which perform well on all training examples. We will extend our discussion of this point to refer to related works.
> >
> > 8) We refer to figure 1c in [2], and figure 5 in [3], for examples where the optimal learning rate is independent of batch size in the large batch limit. See also figure 8 in [4].
> >
> > 9) Yes, this is correct. We use a small but finite learning rate to mean that terms at $O(\epsilon^2)$ are non-negligible while terms at $O(\epsilon^3)$ remain small, and we will clarify this in the text as suggested.
> >
> > Minor comments: We will resolve the issues raised with the references and formatting.
> >
> > Please let us know if you have any further comments or questions.
> >
> > Best wishes,
> > The Authors
> >
> > [1] Stochastic modified equations and adaptive stochastic gradient algorithms, Li et al.
> >
> > [2] On the Generalization Benefit of Noise in Stochastic Gradient Descent, Smith et al.
> >
> > [3] An Empirical Model of Large-Batch Training, McClandish et al.
> >
> > [4] Measuring the Effects of Data Parallelism on Neural Network Training, Shallue et al.

---

> > > ### Comment · AnonReviewer1 · 2020-11-17
> > > **Thanks for the reply**
> > >
> > > Dear authors,
> > >
> > > thanks for the detailed reply. While you have answered some of my questions, several points remain unclear to me. So, I'd like to ask another round of clarifying questions. I'm maintaining the same numbering of the issues and you can consider the points I'm not replying to as resolved.
> > >
> > > 1. After reading your response, I agree that analyzing the mean behavior is a solid contribution and a good first step, but I think this restriction should be made a bit clearer to the reader. However, you didn't address my main question here; to quote myself: *The analysis only guarantees that, from any given starting point $\omega_0$ , the expected iterate after one epoch of SGD ends up close to the ODE path starting from $\omega_0$ . Unless I am missing something, this does not imply that two epochs of SGD starting from $\omega_0$ end up on that path. We can not simply chain two epochs together: The first epoch only stays on the path in expectation, but any realization of that random variable will deviate from the path, which affects the initial condition of the next epoch. Intuitively, one needs to get a handle on the variance of the iterate as well in order to give guarantees for multiple epochs. Is this understanding correct? If so, to what extent can insights about a single epoch of SGD be transferred to practical settings?* I presume the answer is that this type of analysis only describes the local behavior anyway due to the restrictions discussed in point 4, is that fair? In any case, the (non-)transfer to multiple epochs should be mentioned in the paper!
> > >
> > > 2. -
> > >
> > > 3. Personally, I would give the result in Eq. (22) directly, but I understand your arguments for doing it in two stages.
> > >
> > > 4. You give good justifications for the approximation here. These should be in the paper! In particular something along the lines of *"We derive the lowest order correction to gradient flow for SGD with finite learning rate."*
> > >
> > > 5. This is somewhat surprising to me. I'm not aware of any prior work that suggests a qualitative difference **in terms of generalization** between random sampling without replacement and the shuffle-and-traverse strategy. Are you? Or is this just a technical limitation of this type of analysis? This might be a somewhat too technical discussion for this forum, but I don't see where things go wrong when we would go through Equations (12)--(17) and replace the fixed-composition batches with randomly-sampled ones? Shouldn't we arrive at a result similar to Eq. (22)??
> > >
> > > 6. I have probably phrased my question ambiguously. My main point was that *"$m\epsilon$ is small"* can not be the full story. The scale of $m\epsilon$ should somehow be related to some regularity of the objective in order to guarantee that the approximation is good, no? To quote myself again: *"Any given loss function can be rescaled by a constant $M\gg 1$ while scaling the step size with $1/M$. This leaves the behavior of SGD unaffected while making the step size arbitrarily small. Why does that not enter into the analysis?"* I might be missing something obvious here, but this seems like a crucial point to me. Can you clarify this?
> > >
> > > 7. Agreed that understanding these terms can be deferred to future work, but I would maybe be a bit more careful with calling it a "regularizer" then. Just a quick comment: The intuition you've outlined doesn't seem compelling to me. The batch gradient norm is unrelated to "sharpness"; any local minimum of the batch loss has zero batch gradient.
> > >
> > > Edit: Fixing the enumeration.

---

> > > > ### Author Response · Authors · 2020-11-18
> > > > **Reply part 1**
> > > >
> > > > Thank you for your rapid reply! We really appreciate the level of detail in your review.
> > > >
> > > > We clarify our responses below. Please feel free to message us again if anything remains unclear. We are currently working on an updated version of the paper incorporating your and the other reviewers’ comments, which we will upload before the end of the rebuttal period.
> > > >
> > > > 1. We will make clear in the updated version of the paper that we analyze the mean behaviour of the iterates, and discuss this in more depth.
> > > >
> > > >  We apologize for not directly addressing the question of whether we can chain together multiple epochs directly in our first response. As you suggest in your reply, the answer to this question is a little subtle (and interacts with our response to point 4):
> > > >
> > > >  - We can repeat our analysis over any finite number of epochs $n$ by taking a Taylor expansion in terms of $O(nm\epsilon)$ rather than $O(m\epsilon)$. For any number of epochs, the lowest order correction to the mean SGD iterate will lie on the modified loss (equation 1), where the implicit regularization term is an average over all minibatches sampled during the $n$ epochs.
> > > >
> > > >  - However, the Taylor expansion above requires $\epsilon < \epsilon_{max}$ where $\epsilon_{max} \propto 1/n$, and therefore for large $n$ the analysis above would not hold for practical learning rates.
> > > >
> > > >  To summarize, our analysis does identify the bias in the lowest order correction to the dynamics over multiple epochs, however the more epochs we consider the more likely higher order terms are to contribute to the dynamics, and consequently our analysis is most accurate in practice locally over a small number of epochs. We will clarify this point in the text.
> > > >
> > > >  Our experimental results suggest that studying the mean SGD iterate over one epoch up to $O(\epsilon^2)$ can explain most of the generalization benefit of finite learning rate SGD over multiple epochs, at least for Wide-ResNets/CIFAR-10. However, we agree that higher order terms and the variance in the updates may play a role in other practical problems.
> > > >
> > > > 4. Yes, we agree. We intend to extend the discussion of the assumption that $m\epsilon$ is small when we update the text to incorporate these points.
> > > >
> > > > 5. This is a very subtle point, which we hope we can clarify below. (We assume you mean random sampling with replacement, instead of without, in your most recent comment.)
> > > >
> > > >  - To see why random sampling changes our analysis, look at equations 15 and 16. If the minibatches are randomly sampled with replacement, then in equation 15 there would be an expectation on the RHS over the composition of the minibatches. Since the minibatches $j$ and $k$ are now random and independent, the RHS of equation 16 simplifies to $(m^2 - m)/2 \nabla \nabla C \nabla C$. Crucially, the term in $\nabla \nabla \hat{C_j} \nabla \hat{C}_j$ vanishes. In addition, random shuffling will of course also introduce a source of variance at $O(\epsilon)$.
> > > >
> > > >  - Consequently, for random sampling the dynamics up to $O(\epsilon^2)$ are described by the modified stochastic differential equation of equation 7 in [1].
> > > >
> > > >  - However, we are not claiming that there is a qualitative difference in terms of generalization between the random sampling and the random shuffling cases. We believe that in practice this SDE may often have a very similar bias to the ODE identified in our work. Our intuition is that the noise source in the SDE, although it appears as a variance term at $O(\epsilon)$, could implicitly introduce a bias at $O(\epsilon^2)$ when integrated over a finite timescale.
> > > >
> > > >  - The main point we were trying to make in our original response is that, although random sampling and random shuffling may behave similarly in practice, it is much easier to interpret the dynamics for random shuffling, because the mean evolution is described by an ODE, whereas for random sampling we have to study the properties of an SDE, which is more challenging.
> > > >
> > > >  - In addition, random shuffling is more commonly used in practice, and it has better convergence properties since it suppresses the variance at $O(\epsilon)$ [2]. In practice, we found on our Wide-ResNet/CIFAR-10 setup that random shuffling achieves slightly higher test accuracies than random sampling.

---

> > > > > ### Author Response · Authors · 2020-11-18
> > > > > **Reply part 2**
> > > > >
> > > > > 6. Note that the $n$-th order terms in the Taylor expansion of the modified flow have the form $\epsilon^n f_{n}(C(\omega))$, where $f_{n}(C(\omega))$ is a function of the loss $C(\omega)$. When scaling $C$ by $M$, one would obtain $f_{n}(MC(\omega)) = M^{n+1} f_{n}(C(\omega))$. As an example of this, if you inspect equation 9 in the text, we find $f_1(MC(\omega)) = (1/2) \nabla \nabla MC(\omega) \nabla MC(\omega) = M^2 f_1(C(\omega))$.
> > > > >
> > > > >  Consequently if we scale the loss by a factor $M$ and divide the learning rate by the same factor, then the $n$-th order correction term $(\epsilon/M)^n f_n(MC(\omega)) = M \epsilon^n f_n(C(\omega)$ also increases by a factor of $M$. Therefore if we scale the loss by $M$ and the learning rate by $1/M$ then the modified flow will also scale by a factor of $M$, and therefore the dynamics of the parameters are unaffected as required.
> > > > >
> > > > >  The functions $f_{n}(C(\omega))$ are composed of derivatives of the loss $C(\omega)$, therefore these derivatives must also be bounded, and this is implicit in our analysis. A standard result in backward error analysis [3] shows that the smoother the original vector field (i.e., the larger the ball about the current parameters for which the gradient of the loss is analytic), the larger the learning rate for which backward error analysis will be accurate. We will clarify this point in the updated text.
> > > > >
> > > > > 7. We call the first order correction a regularizer since it appears as an additive term in the modified loss with a tunable coefficient (the learning rate), and because tuning the scale of this term enhances the test accuracy (at least for some tasks/datasets). We note that it is common in deep learning to use the word regularizer in a fairly loose sense. For instance, early stopping and batch normalization are commonly referred to as sources of regularization.
> > > > >
> > > > >  While the norm of the full batch gradient vanishes at local minima it does not vanish in the vicinity of local minima, and consequently it may introduce a dynamical penalty favouring flatter minima. However, we agree that this intuition, while interesting, is not compelling.
> > > > >
> > > > > [1] Stochastic modified equations and adaptive stochastic gradient algorithms, Li et al.
> > > > >
> > > > > [2] Random Shuffling Beats SGD after Finite Epochs, Haochen and Sra.
> > > > >
> > > > > [3] Geometric numerical integration: structure-preserving algorithms for ordinary differential equations, Hairer et al.

---

> > > > > > ### Comment · AnonReviewer1 · 2020-11-19
> > > > > > **Reviewer response**
> > > > > >
> > > > > > Thanks for the in-depth response. This clarifies almost all of the open questions I had. I still can't fully wrap my head around the discrepancy between random sampling and shuffle-and-traverse. But irrespective of that, I've become convinced that this analysis provides many interesting insights and I will raise my score and **recommend accepting the paper.** Please implement the changes/clarifications as indicated in your responses!

---

> > > > > > > ### Author Response · Authors · 2020-11-20
> > > > > > > **Author response**
> > > > > > >
> > > > > > > Thank you! We are really grateful for the effort you have put into reviewing our work.
> > > > > > >
> > > > > > > We are working to make the changes in the text as promised.

---

### Official Review · AnonReviewer4 · 2020-10-29
**Borderline Accept Paper**

**Rating:** 7
**Confidence:** 3

**Review:**

Summary: To analyze why the generalization error of SGD with larger learning rates achieves better test error, this paper analyzes the implicit regularization of SGD (with a finite step size) via a first order backward error analysis. Under this analysis the paper shows that the mean position of SGD with $m$ minibatches effectively follows the flow according to Eq (20) for a small but finite step size, while GD effectively follows the last inline equation in section 2.1. The paper shows empirically on an image classification task that by explicitly including the (implicit SGD) regularizer, SGD on the modified loss behaves similarly to using a larger learning rate when evaluating on the test set. The paper then extends this results to consider varying the batch size in section 3, showing that for small batchsizes the implicit regularization scales with the ratio of learning rate and batchsize $\epsilon/B$. Finally in section 4, the paper analyzes SGD when for each sampled minibatch in an epoch, we apply $n$ gradient steps with a stepsize $\epsilon/n$ and show that performance degrades as $n$ increases, suggesting that the benefit of SGD with larger learning rates is due to the implicit regularizer and not the temperature of an associated SDE.

This paper is clearly written and well edited. I find the main result and the analysis technique interesting and novel. Although the experiments are well explained and help support the theory developed, there is only one experiment setting making it difficult to believe strong general claims such as those in section 4. I do have concerns about equating the "mean" behavior of SGD with the actual behavior of SGD and.

Recommendation:
I recommend accepting this paper. As it currently stands, this paper is borderlin on the acceptance threshold for me. I like the novel use of the backward error analysis to gain insight into the behavior of SGD and I believe it would be of interest to ICLR readers. My main concerns are the papers' narrow focus on the mean behavior of SGD and the single experiment setting used to validate results. I would much more strongly support this paper if the theoretical analysis was stronger (e.g. analyzing the variance of individual SGD flows/regularizers to the mean SGD flow/regularizer) or if more experiments (in different settings) supported the results.

Questions:
If we don't take the expectation over $\xi(m)$ in Section 2.2, the theory suggests that there exist a (random) modified flow for each (random) ordering of minibatches $\hat{C}_0, \ldots, \hat{C}_m$ by equating equations (14) and (19). The main result Eq (20) would correspond to the expected value over the (random) modified flow. I believe this paper would be much stronger if there was some discussion of how the variance / deviations of these random flows from the mean flow (i.e the variance of $\xi(m)$) affects the implicit regularization and how this scales with batch size and properties of the loss. Would the implicit regularization break down for some experiments?

Is the assumption that $m \epsilon$ is small reasonable (so that we can ignore the higher order $O(m^3 \epsilon^3)$ terms in the analysis)? Isn't $m = N/B$ the number of updates per epochs very large in practice since $N >> B$?

---

> ### Author Response · Authors · 2020-11-16
> **Thank you for your review**
>
> Thank you for your helpful feedback, and your recommendation to accept our work.
>
> Your review raises three important points, which we discuss below. We are currently working on an updated manuscript to incorporate your comments, and we believe that this will significantly improve the paper.
>
> **Additional experiments:**
> We apologize that the original submission appears to make overly strong empirical claims in section 4. This was not our intention. Our goal was simply to provide intriguing evidence that the SDE analogy does not capture the generalization benefit of SGD in at least one case (Wide-ResNets/CIFAR-10). We will update the language in this section to resolve this issue.
>
> We are also currently preparing additional experiments on MNIST using a fully connected network. We hope to add these to the appendices before the end of the rebuttal period. If they are not complete in time we will add them to the final version.
>
> **Bias and Variance:**
>
> Our analysis focuses on the mean evolution of the SGD iterates. The reviewer is correct to comment that we do not study the variance of the iterates, and this is an important point. We will update the text to make sure that this is explained clearly.
>
> We note that, under the random sampling strategy (common in previous theoretical analyses but rarely used in practice), finite learning rates introduce variance in the evolution at $O(\epsilon)$, while the bias arises at $O(\epsilon^2)$, so it is natural to study the variance in this setting. However under the random shuffling strategy we consider, both the bias and the variance arise at $O(\epsilon^2)$. We therefore anticipate that the bias will play a more important role for the random shuffling strategy than is commonly supposed, which was one of the key inspirations for our analysis. Our experiments suggest that the bias in the iterates up to $O(\epsilon^2)$ can explain most of the generalization benefit of finite learning rates, at least for Wide-ResNets/CIFAR-10.
>
> Intriguingly, there is a simple strategy for sampling minibatches which entirely suppresses the variance in the SGD iterates at $O(\epsilon^2)$, while leaving the bias at $O(\epsilon^2)$ unchanged. To achieve this, we shuffle the dataset, perform one epoch of updates, then reverse the dataset and perform a second epoch, iterating through the same minibatches but in the opposite order. We then shuffle again and repeat. If you inspect equations 13-15 in our analysis, you will see that reversing the sequence of the minibatches every second epoch has the same effect as taking the expectation over all possible sequences (it replaces the sum over $k < j$ by a sum over $k \neq j$). Under this "reverse epoch" strategy the path taken by the modified flow will coincide exactly with the discrete iterates of SGD (for a specific training run, at the end of every second epoch).
>
> **Is the approximation reasonable:**
> Our analysis assumes that $m\epsilon = N\epsilon/B$ is small, in order to neglect terms at $O(m^3\epsilon^3)$. We agree that this is at first sight an extreme approximation and that, as we already note in the text, higher order terms are also likely to play a role in practice. We anticipate that these terms could be an interesting avenue for future work, and we will make this point clearer in the updated text. However, we believe the approximation is reasonable for the following reasons:
>
> 1) Our work identifies the lowest order correction to gradient flow for SGD with finite learning rates. We believe that this is in itself a significant theoretical contribution, especially since this lowest order correction has a natural interpretation as an implicit regularizer.
>
> 2) Our experiments suggest that, at least for Wide-ResNets/CIFAR-10, our analysis is sufficient to describe most of the generalization benefit of large learning rates.
>
> 3) Furthermore, as we show in section 3, the optimal learning rate is usually proportional to the batch size for small batch sizes, and therefore N\epsilon/B does not grow as the batch size falls. Consequently, our approximation may be surprisingly accurate in practice, even when the number of updates in an epoch is large.
>
> 4) Many previous well-cited works in this area make even more extreme approximations, yet these works have still yielded useful insights. For instance, the SDE analogy neglects all terms of $O(\epsilon^2)$ and above, while our analysis only neglects terms at $O(\epsilon^3)$.
>
> We will update the text to clarify all of the points above. Please let us know if there are any other issues you would like us to address.
>
> Best wishes,
> The Authors

---

> > ### Comment · AnonReviewer4 · 2020-11-19
> > **Questions to the response.**
> >
> > For the approximation $m\epsilon = N\epsilon/B$, I agree that $\epsilon/B$ is typically kept constant (e.g. large batch sizes $B$ allow for larger step sizes $\epsilon$), but wouldn't $m$ still increase as $N$ increases (while $B$ and $\epsilon$ are held fixed)? Doesn't your analysis neglects terms at $O(m^3 \epsilon^3)$ instead of $O(\epsilon^3)$? The first 2 reasons in your response are perhaps motivate why the results are useful (regardless), but I don't understand why I shouldn't expect things to breakdown for large $m$.
> >
> > Did you have any thoughts of what conditions we might expect the implicit regularization / low order flow correction to break down? I agree that the implicit regularization appears to explain a lot of the benefit in your experiment settings, but I am hoping to understand / let other readers know what conditions, where SGD + GD with your regularization may differ (e.g. when the variance is not controlled or when $m$ is large).  It may be the case that SGD does not perform as well as GD + regularization when the approximation breaks down, which would further support this implicit regularization as being helpful.
> >
> > Finally, the "reverse epoch" strategy paragraph in the response does not allow the flow to "coincide exactly" with discrete SGD because it's neglecting higher order terms $O(m^3 \epsilon^3)$, right? (Or only coincide up to $O(m^3 \epsilon^3)$ terms). The actual trace of SGD would have the $\omega_k$ varying at each step.

---

> > > ### Author Response · Authors · 2020-11-20
> > > **Additional clarifications**
> > >
> > > Thank you for your reply!
> > >
> > > We respond to your questions below. Please let us know if anything is unclear or if there are other points you would like to discuss.
> > >
> > > **Is the approximation $m \epsilon$ is small reasonable for large datasets:**
> > >
> > > - Yes, our analysis neglects terms at $O(m^3\epsilon^3)$. However in our main analysis in section 2.2, we assume the dataset size $N$ and the batch size $B$ are fixed, and consequently $m =N/B$ is a constant. In this context we can equivalently say that we neglect terms at $O(\epsilon^3)$.
> > >
> > > - If the dataset size is not fixed, the accuracy of the approximation that $m \epsilon = N \epsilon/B$ is small may degrade as $N$ increases, on the basis of the reasonable assumption that the ratio $\epsilon/B$ which achieves the optimal test accuracy is constant as $N$ rises.
> > >
> > > - Interestingly, we note that prior work [1] found evidence to suggest that for SGD without learning rate decay, if we want to maximize the test accuracy, then we should keep $N\epsilon/B$ constant as N increases (i.e., when the dataset size increase the optimal value of the ratio $\epsilon / B$ falls). If this were the case, then the accuracy of our approximation would not fall as $N$ increases, but we note that [1] only shows results for a simple MLP on MNIST and we are not aware of any follow up work which verifies this claim on larger models.
> > >
> > > - To summarize, we agree that it is plausible that the accuracy of our approximations may degrade as the size of the dataset grows, and we will clarify this in the text. However there is also some empirical evidence to suggest this may not occur in practice.
> > >
> > > **When do we expect our analysis to break down:**
> > >
> > > There are three main situations in which we might expect our analysis to break down:
> > >
> > > 1) The variance in the SGD iterates at $O(m^2 \epsilon^2)$ may be large. In this case, while the mean SGD iterate over one epoch will coincide with our analysis, it may not be a good estimate of the iterates of any specific SGD run. (Note however that we can resolve this failure mode using the reverse epoch strategy).
> > >
> > > 2) If $m\epsilon$ is too large, then higher order terms will also contribute to the dynamics.
> > >
> > > 3) There may be situations in which these higher order terms, while small over a single epoch, accumulate over many epochs and cause the mean SGD iterate to slowly diverge from the path of the modified loss on long timescales (even though the SGD iterates in any single epoch stay close to the path of the modified loss, conditioned on the initial parameters of that epoch).
> > >
> > > We anticipate that situation 3 will be quite common in practice; i.e., the dynamics of SGD at practical learning rates will not exactly coincide with the modified loss over an entire training run, however the modified loss will provide a good approximation to the path of SGD locally over a few epochs, and consequently our analysis will still provide useful insights into the implicit regularization phenomenon, as shown by our experiments.
> > >
> > > **Reverse epoch strategy:**
> > >
> > > Our apologies. Yes, we meant to say that the reverse epoch strategy coincides exactly with gradient flow on the modified loss for terms below $O(m^3 \epsilon^3)$, because it eliminates the variance at $O(m^2 \epsilon^2)$. However it does not exactly match the modified flow for higher order terms, and therefore this equivalence only holds if $m \epsilon$ is sufficiently small.
> > >
> > > Best wishes,
> > > The Authors
> > >
> > > [1] A Bayesian Perspective on Generalization and SGD, Smith and Le

---

### Official Review · AnonReviewer2 · 2020-10-29
**Good paper, can be improved by varying experiments**

**Rating:** 7
**Confidence:** 3

**Review:**

This paper studies an implicit regularization mechanism of finite learning rate SGD by introducing explicitely a regularization term, using the framework of backward analysis.

They theoretically motivate their analysis, then empirically demonstrate it on CIFAR-10 using a Wide ResNet architecture.

This extends a previous (Barrett and Dherin, preprint) analysis of GD using the same framework, but limited to full batch GD. Noticeably, this new analysis using minibatch GD highlights an additional regularization of the trace of the covariance of per-example gradients.

In sec 2, however, I think it should be made clear that the setup is slightly different from minibatch GD, even when trained for a single epoch, in that there is an expectation accross permutations of sequences of minibatches. Can you discuss this assumption a bit more?

In terms of experiments, it would be useful to include other architecture/tasks, even toyish, in order to appreciate the generality of the empirical evaluation.

Overall, I think this contributes new interesting insights which are very relevant for studying minibatch GD in deep learning.

---

> ### Author Response · Authors · 2020-11-16
> **Thank you for your review**
>
> Thank you for your positive feedback and useful comments!
>
> **Bias and Variance:**
> You raise an important point. Our analysis in section 2 focuses on the mean evolution of SGD, averaged over all possible sequences of a fixed set of minibatches. It therefore identifies the bias that arises from finite step sizes, but neglects the variance in the iterates across different training runs. To expand further on this point:
>
> Under random sampling strategies (often analyzed in previous theoretical work, but rarely used in practice), SGD exhibits variance at $O(\epsilon)$, but the bias from finite learning rates only arises at $O(\epsilon^2)$. It is therefore natural to focus on the variance rather than the bias in the limit of small learning rates. However under the random shuffling strategy we study in this work (which is also more common in practice), both bias and variance arise at $O(\epsilon^2)$. We therefore anticipate that the bias will play a more important role for the random shuffling strategy than is commonly supposed, which was one of the key inspirations for our analysis. Our experiments suggest that the bias in the iterates up to $O(\epsilon^2)$ can explain the generalization benefit of finite learning rate SGD, at least for Wide-ResNets/CIFAR-10.
>
> Intriguingly, we can construct a specific sequence of minibatches which further suppresses the variance, such that the bias at $O(\epsilon^2)$ is unchanged but the variance arises only at $O(\epsilon^3)$. To do this, we shuffle the dataset, perform a single epoch, then reverse the dataset and perform a second epoch, iterating through the same sequence of minibatches but in the opposite order. We then shuffle the dataset and repeat. If you inspect equations 13-15 in our analysis, you will see that reversing the sequence of the minibatches on every second epoch has the same effect as taking the expectation over all possible sequences (it replaces the sum over $k < j$ by a sum over $k \neq j$). Under this "reverse epoch" strategy the path taken by the modified flow will coincide exactly with the discrete iterates of SGD (for a specific training run, at the end of every second epoch).
>
> We will update the text to ensure that it is clear that our analysis focuses on the mean evolution. We will also describe the "reverse epoch" strategy above, and extend our discussion of the relative roles of bias and variance under different sampling strategies.
>
> **Additional experiments:**
> We are currently preparing additional experiments on MNIST using a fully connected network. We hope to add these to the appendices before the end of the rebuttal period, and if they are not complete in time we will add them to the final version.
>
> We hope that the clarifications above are helpful. Please let us know if you have any more questions or comments.
>
> Best wishes,
> The Authors

---

### Official Review · AnonReviewer3 · 2020-11-02
**Review of On the Origin of Implicit Regularization in Stochastic Gradient Descent**

**Rating:** 8
**Confidence:** 3

**Review:**

This paper analyzes the implicit regularization in SGD with finite learning rates via backward error analysis. The modified flow introduced in this paper better approximates the practical behavior of SGD as it does not require vanishing learning rates and it allows to use random shuffling in stead of i.i.d sampling. The numerical experiments validates the existence of the implicit regularization and how it affects the generalization of the model trained by SGD. The difference from SDE analysis is also discussed.

Reason for score:
1. The paper is well organized. Specially, I enjoy reading section II. The tool of backward error analysis and the derivation of the implicit regularization in SGD flow are introduced clearly and concisely. The analysis is based on random shuffling instead of i.i.d sampling matches the practical use of SGD.
2. The numerical experiments are very convincing. The consistency of SGD with larger lr and SGD with smaller lr plus explicit regularization validates the results of theoretical analysis. The numerical experiments also provide some insights into tuning hyper parameters such as learning rate and batch size.

---

> ### Author Response · Authors · 2020-11-16
> **Thank you for your review**
>
> Thank you for your positive feedback. We are really glad that you enjoyed reading the paper!
> Please let us know if you have any further questions.
>
> Best wishes,
> The Authors

---

### Author Response · Authors · 2020-11-24
**We have updated the manuscript**

We thank the reviewers again for their insightful comments. We have now uploaded an updated version of the paper to openreview which incorporates most of the comments raised. We believe that this has significantly improved the paper. To summarize, the main changes in the new version are:

1. We have added a new subsection in section 2.2 (titled “Remarks on the Analysis”) which discusses the assumptions made in our analysis in more depth. Specifically, we clarify the following:
 - We make clear that we study the mean evolution of the iterates, not the variance of individual training runs. We discuss why we make this choice by comparing the roles of bias and variance in the random sampling strategy and the random shuffling strategy.
 - We clarify that the “reverse-epoch strategy” can enable us to suppress the variance of individual training runs at $O(\epsilon^2)$.
 - We provide a discussion of our assumption that terms of $O(m^3 \epsilon^3)$ are small and its implications.


2. We have added a set of new experimental results on a 3-layer MLP classifying Fashion-MNIST in appendix section D. We chose to use Fashion-MNIST as the implicit regularization effect of large learning rates/small batch sizes for MNIST is relatively small. Note that these results are provisional as we currently perform a single training run for each set of hyper-parameters. We will update the figures to provide the average performance over multiple runs in time for the camera ready.


3. We have made several other minor changes throughout the text. To list a few:
 - We clarify that backward error analysis assumes that the original flow is analytic in the vicinity of the parameters.
 - We have softened the language describing our empirical results in section 4.
 - We clarify the precise meaning of the phrase “small but finite learning rates”.
 - We cleaned up the references to refer to published versions of papers.

We will continue working on the text to incorporate the remaining suggestions of the reviewers in time for the camera ready.

Best wishes,
The Authors

---

### Decision · Program_Chairs · 2021-01-07
**Final Decision**

**Decision:**

Accept (Poster)

**Comment:**

Dear authors,

all reviewers found many interesting contributions in your paper and also pointed out some minor/major issues. In your rebuttal discussions, you addressed most of them to their satisfaction and I hope you will incorporate them carefully also in your final submission.

I hence recommend accepting this paper